# DISSECTING CAUSAL BIASES

## ABSTRACT

Accurately measuring discrimination in machine learning-based automated decision systems is required to address the vital issue of fairness between subpopulations and/or individuals. Any bias in measuring discrimination can lead to either amplification or underestimation of the true value of discrimination. This paper focuses on a class of bias originating in the way training data is generated and/or collected. We call such class causal biases and use tools from the field of causality to formally define and analyze such biases. Four sources of bias are considered, namely, confounding, selection, measurement, and interaction. The main contribution of this paper is to provide, for each source of bias, a closed-form expression in terms of the model parameters. This makes it possible to analyze the behavior of each source of bias, in particular, in which cases they are absent and in which other cases they are maximized. We hope that the provided characterizations help the community better understand the sources of bias in machine learning applications.

## 1 INTRODUCTION

Machine learning (ML) is being used to inform decisions with critical consequences on human lives such as job hiring, college admission, loan granting, and criminal risk assessment. Unfortunately, these automated decision systems have been found to consistently discriminate against certain individuals or sub-populations, typically minorities Angwin et al. (2016); Buolamwini & Gebru (2018); O'Neill (2016); Quick (2015); Obermeyer et al. (2019). Addressing the problem of discrimination involves two main tasks. First, measuring discrimination as accurately and reliably as possible. Second, mitigating discrimination. The first task is clearly a prerequisite for the appropriate implementation of the second task. Proposing mitigation policies on the ground that discrimination is significant while it is in fact less significant (let alone not existing) may lead to undesirable consequences.

In this paper, we make a distinction between discrimination and bias. We use the term discrimination to refer to *the unjust or prejudicial treatment of different categories of people, on the ground of race, age, gender, disability, religion, political belief, etc.*. Whereas the term bias is used to refer to *the deviation of the expected value from the quantity it estimates.*

Discrimination in ML decisions can originate from several types of bias as described in the literature. For instance, The Centre for Evidence-Based Medicine (CEBM) at the University of Oxford is maintaining a list of 62 different sources of bias of Oxford (2021). More related to ML, Mehrabi et al. Mehrabi et al. (2021) classify the sources of bias into three categories depending on when the bias is introduced in the automated decision loop. In this paper we focus on a class of biases, we call causal biases, which arise from the way data is generated and/or collected. We use tools from the field of causality Pearl (2009); Imbens & Rubin (2015) as the latter emerged as a way to reliably estimate the effects between variables in presence of data imbalance leading to a deviation between the population distribution and the training data distribution.

The main contribution of the paper is to use tools and existing results from the field of causality to generate closed-form expressions of four sources of bias. These sources of biases correspond to four different causal structures, namely, confounding, colliding (selection), measurement, and interaction. This has at least two advantages. First, understand how bias is expressed in terms of model parameters. Second, analyze the magnitude of each type of bias, in particular, when it is absent and when it is optimal. Finally, we empirically show the extent of causal biases in ML fairness benchmark datasets. All proofs of the closed-form expressions can be found in the supplementary material.

## 2 TYPES OF BIAS

Measuring discrimination without taking into consideration the causal structure underlying the relationships between variables may lead to misleading conclusions. That is, a biased estimation of discrimination. In extreme cases, such as Simpson's paradox, the bias may lead to reversing the conclusions (e.g. the biased estimation indicates a positive discrimination, while the unbiased estimation is actually a negative discrimination).

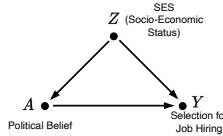
Figure 1: Confounding bias example.

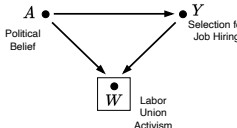
Figure 2: Collider bias example.

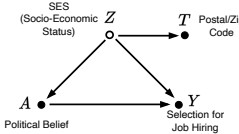
Figure 3: Measurement bias example.

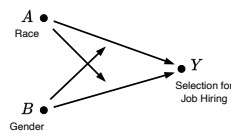
Figure 4: Interaction Bias example.

### 2.1 CONFOUNDING BIAS

The first type of bias, confounding bias, is due to a failure to consider a confounder variable. Consider the hypothetical example in Figure 1 of an automated system to select candidates for job positions. Assume that the system takes as input two features, namely, the socio-economic status (SES) denoted as $Z$ and the political belief of the candidate $A$. The outcome $Y$ is whether the candidate is selected for the next stage of hiring (or the probability the candidate is selected). The outcome $Y$ is influenced by the SES (A better SES makes it possible for candidates to attend more reputable academic institutions and to be enrolled in costly trainings). Both variables can be either binary ($Z$ might be either rich or poor while $A$ might be either liberal or conservative) or continuous (how rich/poor a candidate is for $Z$ and the degree of conservativeness of the candidate for $A$). The political belief $A$ of a candidate can be influenced by several variables, but in this example, assume that it is only influenced by the SES of the candidate. Finally, assume that the automated decision system is suspected to be biased by the political belief of candidates. That is, it is claimed that the system will more likely select candidates with a particular political belief.

A simple approach to check the fairness of the automated selection $Y$ with respect to the sensitive attribute $A$ is to contrast the conditional probabilities: $\mathbb{P}(Y = 1 \mid A = 0)$ and $\mathbb{P}(Y = 1 \mid A = 1)$[1], corresponding to statistical disparity, which quantifies the disparity in the selection rates between both types of candidates (conservatives and liberals). However such estimation of discrimination is biased due to the confounding path through $Z$. As $Z$ variable causes both the sensitive variable $A$ and the outcome $Y$, it creates a correlation between $A$ and $Y$ which is not causal. In other words, high SES (rich) candidates tend to have a more conservative political belief and at the same time more chances to be selected for the job (better academic institutions and training) which creates the following correlation in the data: employers will have more candidates with convervative political beliefs, and hence less candidates with liberal political beliefs. Such correlation is due to the confounder $Z$ and should not count as discrimination. We call such bias in estimating discrimination, confounding bias.

### 2.2 SELECTION BIAS

The second type of bias, selection bias, is due to the presence of common effect (collider) variable and a data generation process implicitly conditioning on that variable. Using the same hypothetical example of job selection, consider the causal graph in Figure 2. $A$ and $Y$ are the same as in the previous example. Assume that data for training the automated decision system is collected from different sources, but mainly from labor union records. Assume also that variable $W$ representing the labor union activism of the candidate is caused by both $A$ and $Y$. On one hand, the political belief $A$ influences whether a candidate is an active member of labor union (individuals with liberal political beliefs are more likely to enroll in labor unions). On the other hand, if a candidate is selected/hired, then there are higher chances that she becomes a member of labor union and consequently that her

---
[1]

case is recorded in the labor union records. Consistent with previous work, a box around a variable ($W$) indicates that data is generated by implicitly conditioning on that variable.

Again the simple approach of contrasting the selection rates between both types of candidates (conservatives and liberals) leads to a biased estimation of discrimination due to the colliding path through $W$. Intuitively, an individual has a record in the collected data either because she has liberal political beliefs or because she is selected for the job. Individuals who happen to have liberal political beliefs and at the same time selected for the job are still present in the data, however conditioning on labor union activism creates a correlation between $A$ and $Y$ which is not causal: data coming from labor union records includes fewer liberal candidates which are selected for the job than conservative candidates. Again, this is a discrimination against candidates with liberal political beliefs. Such correlation is due to the colliding structure and should not count as discrimination. We call such bias in estimating discrimination, selection bias.

## 2.3 MEASUREMENT BIAS

The third type of bias, measurement bias, is due to the use of a proxy variable to estimate discrimination instead of an ideal but unmeasurable variable. Consider a third variant of the same job selection example having the causal graph of Figure 3. Unlike in the causal graph of confounding bias (Figure 1), the confounder variable $Z$ is unmeasurable (empty bullet instead of a filled one). In practice, it is difficult to find a variable that represents accurately the socio-economic status (salary, possessions, etc.). Being unmeasurable, $Z$ cannot be used to estimate discrimination while blocking the confounding path through $Z$. For practical reasons, the (measurable) variable $T$ representing the postal/zip code of the candidate's address can be used instead. $T$ is considered a proxy of $Z$ as it is highly correlated with (but not identical to) $Z$[2]. Using variable $T$ as a proxy to measure $Z$ may lead to an additional bias, we call measurement bias.

## 2.4 INTERACTION BIAS

The fourth type of bias, interaction bias, is observed when two causes of the outcome interact with each other, making the joint effect smaller or greater than the sum of individual effects. Consider the same job hiring example but where two sensitive attributes, political belief (liberals vs conservatives) and gender have an effect on the hiring decision. In the presence of interaction between political belief and gender, statistical disparity will not accurately measure the individual effects of Political Belief and Gender even if no confounding condition is satisfied. For example, it is possible to observe a situation where statistical parity is almost satisfied for both individual sensitive variables, but the intersectional sensitive group is discriminated Buolamwini & Gebru (2018). Following our previous example, we would define liberal females as an unprivileged intersectional group and conservative males as a privileged intersectional group. In the presence of interaction, the discrimination against liberal females is not equal to the sum of discrimination against conservative and females individually. In addition, the average discrimination value for liberals or females, as measured by statistical disparity, will also be biased, as it does not take into account the interaction between the two sensitive variables.

## 3 CONFOUNDING BIAS

Confounding bias occurs when both the sensitive variable and the outcome have a common cause, the counfounder variable. Consequently, the mechanism of selecting samples from the two groups (protected and privileged) is not independent of the outcome. This creates a bias when measuring the causal effect of the sensitive attribute on the outcome.

For a concise notation, let $y_1$ and $y_0$ denote the propositions $Y = 1$ and $Y = 0$, respectively, and the same for the variables $A$ and $Z$. For instance, $\mathbb{P}(Y = 1|A = 0)$ is written simply as $\mathbb{P}(y_1|a_0)$.

**Theorem 3.1.** *Assuming $A$, $Y$, and $Z$ binary variables and that $\mathbb{P}(a_0) = \mathbb{P}(a_1) = \frac{1}{2}$[3], the difference in discrimination due to confounding is equal to:*

$$ConfBias(Y, A) = (1 - \mathbb{P}(z_0|a_0) - \mathbb{P}(z_1))(\alpha - \beta + \gamma - \delta). \quad (1)$$

---

[2]The candidate's address gives a strong indicator of the socio-economic status.

[3]A result without this assumption can be found in the supplementary material.

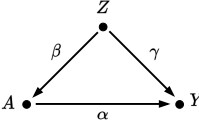

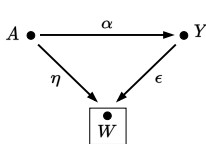

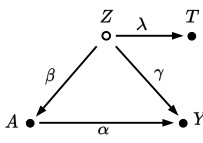

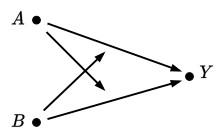

Figure 5: Confounding structure in linear model

Figure 6: Simple collider structure

Figure 7: Simple measurement bias structure

Figure 8: Interaction Bias, where $A$ and $B$ are sensitive variables and $Y$ is an outcome.

**Theorem 3.2.** *Let $A$, $Y$, and $Z$ variables with linear regressions coefficients as in Figure 5 which represents the basic confounding structure. The confounding bias can be expressed in terms of covariances of pairs of variables as follows:*

$$ConfBias(Y, A) = \frac{\sigma_{za}\sigma_{yz} - \frac{\sigma_{ya}}{\sigma_a^2}\sigma_{za}^2}{\sigma_a^2\sigma_z^2 - \sigma_{za}^2} \tag{2}$$

*where $\sigma_x^2$ denotes the variance of $X$ and $\sigma_{xy}$ denotes the covariance of $X$ and $Y$.*

*Confounding bias can also be expressed in terms of the linear regression coefficients as follows:*

$$ConfBias(Y, A) = \frac{\sigma_z^2}{\sigma_a^2}\beta\gamma \tag{3}$$

**Corollary 3.3.** *For standardized variables[4] $A$, $Y$, and $Z$, confounding bias can be expressed in terms of covariances as:*

$$ConfBias(Y, A) = \frac{\sigma_{za}\sigma_{yz} - \sigma_{ya}\sigma_{za}^2}{1 - \sigma_{za}^2} \tag{4}$$

*And in terms of regression coefficient, simply as ( Pearl (2013)):*

$$ConfBias(Y, A) = \beta\gamma \tag{5}$$

Equations (4) and (5) can be obtained from Equations (2) and (3) as $\sigma_z = \sigma_a = 1$.

We provide in the supplementary material (Appendix B.3) the closed-form expression of confounding bias in presence of two (or several confounders) in the linear case.

# 4 SELECTION BIAS

Selection bias occurs when there is collider variable caused by both the sensitive attribute $A$ and the outcome variable $Y$ and the data generation process implicitly conditions on that collider variable. The simplest case is illustrated in Figure 6. Consistent with previous work, a box around a variable indicates that data is generated by conditioning on that variable.

**Theorem 4.1.** *Assuming $A$, $Y$, and $W$ binary variables and that $\mathbb{P}(a_0) = \mathbb{P}(a_1) = \frac{1}{2}$, the difference in discrimination due to collider structure is equal to:*

$$SelBias(Y, A) = (1 - \mathbb{P}(w_0|a_0) - \mathbb{P}(w_1))(-\alpha + \beta - \gamma + \delta) \tag{6}$$

*where, $\alpha = \mathbb{P}(y_1|a_0, w_0)$, $\beta = \mathbb{P}(y_1|a_0, w_1)$, $\gamma = \mathbb{P}(y_1|a_1, w_0)$, and $\delta = \mathbb{P}(y_1|a_1, w_1)$.*

**Theorem 4.2.** *Let $A$, $Y$, and $Z$ variables with linear regressions coefficients as in Figure 6 which represents the basic collider structure. Bias due to selection is equal to:*

$$SelBias(Y, A) = \frac{\frac{\sigma_{ya}}{\sigma_a^2}\sigma_{wa}^2 - \sigma_{wa}\sigma_{yw}}{\sigma_a^2\sigma_w^2 - \sigma_{wa}^2} \tag{7}$$

*Selection bias can also be expressed in terms of the linear regression coefficients as follows:*

$$SelBias(Y, A) = \epsilon \ \frac{\sigma_a^4\alpha^2\eta + \sigma_a^4\alpha^3\epsilon - \sigma_y^2\sigma_a^2\eta - \sigma_y^2\sigma_a^2\alpha\epsilon}{\sigma_a^2\sigma_w^2 - (\sigma_a^2\eta + \sigma_a^2\alpha\epsilon)^2} \tag{8}$$

---

[4]Variables normalized to have a mean 0 and standard deviation 1.

**Corollary 4.3.** *For standardized variables $A$, $Y$, and $W$, selection bias can be expressed in terms of convariances as:*

$$SelBias(Y, A) = \frac{\sigma_{ya}\sigma_{wa}^2 - \sigma_{wa}\sigma_{yw}}{1 - \sigma_{wa}^2} \tag{9}$$

*And in terms of regression coefficient:*

$$SelBias(Y, A) = \epsilon \; \frac{\alpha^2\eta + \alpha^3\epsilon - \eta - \alpha\epsilon}{1 - (\eta + \alpha\epsilon)^2} \tag{10}$$

Equations (9) and (10) can be obtained from Equations (7) and (8) as $\sigma_a = \sigma_w = \sigma_y = 1$.

## 5 MEASUREMENT BIAS

Measurement bias arises from how particular variable(s) are measured. A common example is when the ideal variable for a model is not measurable/observable and instead we rely on a proxy variable which behaves differently in different groups. Figure 7 shows a simple scenario when measuring accurately the discrimination based on $A$ requires adjusting on variable $Z$. However, if $Z$ is not measurable but a proxy variable $T$ is measurable, measurement bias occurs when we adjust on $T$ instead of $Z$.

**Theorem 5.1.** *Assuming $A$, $Y$, and $Z$ binary variables such that $Z$ is not measurable, but only the error mechanism ($\mathbb{P}(T|Z)$) is available, and that $\mathbb{P}(a_0) = \mathbb{P}(a_1) = \frac{1}{2}$, the difference in discrimination due to measurement bias, MeasBias(Y, A) can be expressed in terms of $\mathbb{P}(T|Z)$ as follows:*

$$
\begin{aligned}
&\epsilon(\delta - \beta) + (1 - \epsilon)(\gamma - \alpha) \\
&\quad - \epsilon\big(\delta - \beta + 4\mathbb{P}(t_1|z_0)(\beta - \delta + \gamma\Phi + \gamma\Psi)\big) Q \\
&\quad - (1 - \epsilon)\big(\gamma - \alpha + 4\mathbb{P}(t_0|z_1)(\alpha - \gamma + \delta + \delta\Psi^{-1} + \beta\Phi^{-1})\big) R
\end{aligned} \tag{11}
$$

*where:*

$$
\begin{array}{cccc}
\alpha = \mathbb{P}(y_1|a_0, t_0) & \gamma = \mathbb{P}(y_1|a_1, t_0) & Q = \dfrac{1 - \frac{\mathbb{P}(t_0|z_1)}{\epsilon}}{1 - \frac{\mathbb{P}(t_0|z_1)}{2\epsilon}} & \Phi = \dfrac{\epsilon + \frac{\tau}{2} - 1}{\epsilon + \frac{\tau}{2} - \frac{1}{2}} \quad \epsilon = \mathbb{P}(t_1) \\[3mm]
\beta = \mathbb{P}(y_1|a_0, t_1) & \delta = \mathbb{P}(y_1|a_1, t_1) & R = \dfrac{1 - \frac{\mathbb{P}(t_1|z_0)}{1-\epsilon}}{1 - \frac{\mathbb{P}(t_1|z_0)}{2-2\epsilon}} & \Psi = \dfrac{1 - \tau}{\tau} \qquad \tau = \mathbb{P}(t_0|a_0)
\end{array}
$$

**Theorem 5.2.** *Let $A$, $Y$, $Z$, and $T$ variables with linear regressions coefficients as in Figure 7 which represents the basic measurement bias structure. Bias due to measurement error is equal to:*

$$MeasBias(Y, A) = \frac{\sigma_z^2\beta\gamma(\sigma_t^2 - \sigma_z^2\lambda^2)}{\sigma_a^2\sigma_t^2 - \sigma_z^4\lambda^2\beta^2} \tag{12}$$

**Corollary 5.3.** *For standardized variables $A$, $Y$, $Z$, and $T$, measurement bias is equal to:*

$$MeasBias(Y, A) = \frac{\beta\gamma(1 - \lambda^2)}{1 - \lambda^2\beta^2} \tag{13}$$

## 6 INTERACTION BIAS

Interaction bias takes place in the presence of two sensitive attributes when the value of one sensitive attribute influences the effect of the other sensitive attribute on the outcome (Figure 8).

**Theorem 6.1.** *Under the assumption of no confounders between $A$ and $Y$ on one hand, and between $B$ and $Y$ on the other hand, adding up the single effects of $A$ and $B$ on $Y$ to estimate the discrimination due to both sensitive variables leads to a biased estimation. The amount of the bias ($IntBias$) coincides with the interaction term as follows:*

$$
\begin{aligned}
IntBias(Y, A, B) &= Interaction(A, B) \\
&= P(y_1|a_1, b_1) - P(y_1|a_0, b_1) - P(y_1|a_1, b_0) + P(y_1|a_0, b_0)
\end{aligned} \tag{14}
$$

**Theorem 6.2.** *Under the assumption of no confounders between $A$ and $Y$ on one hand, and between $B$ and $Y$ on the other hand, the interaction bias when estimating discrimination with respect to only $A$ is equal to:*

$$IntBias(Y, A) = P(b_1)Interaction(A, B) \tag{15}$$

*IntBias*$(Y, B)$ is defined similarly as $P(a_1)Interaction(A, B)$. For the linear model case, consider the true model:

$$Y = \beta_0 + \beta_1 A + \beta_2 B + \beta_3 AB \tag{16}$$

and a biased model, that does not include interaction term $\beta_3$:

$$Y = \beta_0' + \beta_1' A + \beta_2' B \tag{17}$$

Where $A$ and $B$ are binary sensitive attributes and $Y$ is a continuous value outcome. The change in Y due to $A$ is $\beta_1 + \beta_3 B$ and, similarly the change in Y due to $B$ is $\beta_2 + \beta_3 A$ Keele & Stevenson (2021). In this case, a measure of effect of $A$ ($\beta_1'$) or $B$ ($\beta_2'$) without an interaction term would be inaccurate.

**Theorem 6.3.** *Let $A,B$ and $Y$ be variables with linear regression coefficients as in Equation 17. In a linear model with binary $A$ and $B$ the bias due to interaction, when measuring the discrimination with respect to both $A$ and $B$ is equal to:*

$$IntBias(Y, A, B) = (\beta_1' + \beta_2') - (\beta_1 + \beta_2 + \beta_3) = \beta_3$$

Intuitively, $\beta_3$ is part of an effect of the intersectional sensitive variable $A_1 B_1$ on $Y$ that is left out of the estimation when fitting linear regression without the interaction term.

**Theorem 6.4.** *Let $A,B$ and $Y$ be variables with linear regression coefficients as in Equation 17. In a linear model with binary $A$ and $B$ the bias due to interaction, when measuring the discrimination with respect to only $A$ is equal to:*

$$IntBias(Y, A) = \beta_3 \mathbb{P}(B_1) \tag{18}$$

Intuitively, *IntBias*$(Y, A)$ measures how wrong is the evaluation of effect of $A = 1$ on average, for cases where $B = 1$ or $B = 0$. Similarly, *IntBias*$(Y, B) = \beta_3 \mathbb{P}(A_1)$.

## 7 BIAS ANALYSIS

Expressing different types of bias in terms of the model parameters (conditional probabilities and regression coefficients) allows to study the behavior of bias and how it is impacted by the different parameters. In particular, at which parameters value it is peaked and at which other values it is absent. The aim is to identify the cases where a given estimation of discrimination is biased and at which extent.

### 7.1 BINARY CASE

**Confounder Bias** is absent when at least one of the two terms of Equation 1 is equal to 0. For the first term ($1 - \mathbb{P}(z_0|a_0) - \mathbb{P}(z_1) = 0$), it is easy to show that it is equivalent to $\mathbb{P}(z_0|a_1) = \mathbb{P}(z_0)$ which is in turn means that $Z$ and $A$ are independent ($A \perp\!\!\!\perp Z$).

The second term is equal to 0 when :

$$\mathbb{P}(y_1|a_0, z_0) - \mathbb{P}(y_1|a_0, z_1) = -(\mathbb{P}(y_1|a_1, z_0) - \mathbb{P}(y_1|a_1, z_1)) \tag{19}$$

Thre right-hand side can be interpreted as the Contolled Direct Effect (CDE) VanderWeele (2011) of $Z$ on $Y$ when $A = 0$ whereas the left-hand side is the opposite of $\mathbb{P}(y_1|a_1, z_0) - \mathbb{P}(y_1|a_1, z_1)$ which is the CDE of $Z$ on $Y$ when $A = 1$. Confounding bias is equal zero, when the CDE of $Z$ on $Y$ when $A = 1$ is the exact opposite of to that when $A = 0$. In the job hiring example of Figure 1, it means that we privilege poor liberals as much as we privilege rich conservatives, therefore the effect $Z-> Y$ is canceled out. Equation 19 can also hold when both sides are equal to 0. This means that $Z$ has no direct effect on $Y$ (no edge between $Z$ and $Y$). $Z$ can still have effect on $Y$ which is mediated through $A$, but it does not have a role as a confounder. To summarize, confounding bias is

absent in three cases: either $A \perp\!\!\!\perp Z$ ($A$ and $Z$ are independent) or the edge $Z \to Y$ is absent, or the CDE of $Z$ on $Y$ when $A = 0$ and $A = 1$ are opposite and hence cancel each others.

Confounding bias is peaked when the first term $(1 - \mathbb{P}(z_0|a_0) - \mathbb{P}(z_1))$ is equal to 1 or $-1$ *and* the second term $(-\alpha + \beta - \gamma + \delta)$ is equal to 2 or $-2$. The first term is equal to 1 when $\mathbb{P}(z_1) = 0$ and $\mathbb{P}(z_0|a_0) = 0$. This is an extreme situation when all data instances have the same values of $A$ and $Z$ variables, that is, $a_1$ and $z_0$. The same term is equal to $-1$ when $\mathbb{P}(z_1) = 1$ and $\mathbb{P}(z_0|a_0) = 1$ which corresponds to the other extreme situation of all data instances have $a_0$ and $z_0$. In the job hiring example, both cases correspond to a situation when all candidates are of the same type: poor liberals or rich liberals. The second term reaches a peak value (2.0 or $-2.0$) when the CDE of $Z$ on $Y$ is maximum (1 or $-1$) for both $a_0$ and $a_1$. To summarize, confounding bias is optimal when the effect through the edge $Z \to A$ is very strong (first term) *and* the effect through the edge $Z \to Y$ is very strong (second term). This optimal situation can be seen as an extreme case of Simpson's paradox Simpson (1951).

**Collider Bias** Collider bias can be viewed as an inverse case of a confounder bias. While confounder bias compromises internal validity, selection bias is a threat to external validity Haneuse (2016). Similarly as confounder bias, collider bias does not manifest if the direct link between $A$ and $W$ or $Y$ and $W$ is absent, or the link between $W$ and $Y$ is the opposite for the values $A = 1$ and $A = 0$. The bias is maximized when the group corresponding to $A = 1$ and $W = 0$ is very large (the negative bias case would occur if the group $A = 1$ and $W = 1$ is dominant). Maximization of bias also requires that the link from $Y$ to $W$ is deterministic and has the same direction for both values of $A$.

**Measurement Bias** depends heavily on $\mathbb{P}(T|Z)$. For instance, from Theorem 5.1, it is easy to show that if $\mathbb{P}(t_0|z_1) = \mathbb{P}(t_1|z_0) = 0$ ($T$ and $Z$ are fully dependent), then $Q = R = 1$, and consequently measurement bias disappears. Conversely, if $\mathbb{P}(t_0|z_1) = \mathbb{P}(t_1) = \epsilon$ and $\mathbb{P}(t_1|z_0) = \mathbb{P}(t_0) = 1 - \epsilon$ ($T$ and $Z$ are independent), then $Q = R = 0$, and consequently, measurement bias is maximized as the two negative terms of Equation equation 11 disappear. The maximum value of measurement bias in that case is $\epsilon(\delta - \beta) + (1 - \epsilon)(\gamma - \alpha)$.

**Interaction Bias** Interaction bias for the intersectional case coincides with the interaction term. More precisely, it is maximized when the interaction is maximized and diminishes when the interaction is small. Note that the interaction is equal 0 when one of the sensitive attributes does not have an effect on $Y$ Rothman et al. (2008). The interaction bias when measuring the effect of one sensitive attribute $A$ or $B$ on $Y$ depends on the interaction term and the probability of $B = 1$ and $A = 1$, respectively. The bias increases with the probability of $A = 1$ or $B = 1$ and the interaction term. Interaction bias is equal to zero when either interaction, to the probability of $B = 1$ or $A = 1$, respectively, is equal to 0.

## 7.2 LINEAR CASE

To analyze the different types of bias in the linear case, we generate synthetic data according to the following models. Without loss of generality, the range of possible values of all coefficients ($\alpha, \beta, \gamma, \eta, \epsilon$, and $\delta$) is $[-1.0, 1.0]$ (Appendix D.2). Figure 9 shows the magnitude of each type of bias based on the expressions obtained in Sections 3, 4, and 5. In particular, Equations 3 for confounding bias, 8 for selection bias, and 12 for measurement bias. Three dimensions plot is used for confounding bias (Figure 9(a)) as bias is expressed in terms of two variables ($\beta$ and $\gamma$) whereas four dimensions plots are used for selection and measurement biases (three variables). Confounding bias is maximized when both $\beta$ and $\gamma$ have extreme values ($+1.0$ or $-1.0$): positive bias when $\beta$ and $\gamma$ are of the same sign, and negative otherwise. Bias is absent when at least one of the coefficients is zero. In between these extreme cases, confounding bias has strictly linear relation with $\beta$ whereas a non-linear relation with $\gamma$. More importantly, confounding bias is more sensitive to $\beta$ than to $\gamma$ particularly for extreme values (when coefficients are close to $+1.0$ or $-1.0$). That is, modifying the effect of the confounder (e.g. $Z$) on the sensitive variable (e.g. $A$) has more impact on the confounding bias than modifying the effect of the confounder on the outcome variable (e.g. $Y$) with the same amount. In the job hiring example (Section 2.1) this means that the effect of Socio-Economic status on political belief has more impact on the counfounding bias than the effect of socio-economic status on job hiring. However, if the variables are standardized, both effects contribute equally to confounding bias (Corollary 3.3).

Unlike confounding bias, the magnitude of selection bias (Figure 9(b)) depends also on the regression coefficient of $Y$ on $A$ ($\alpha$). Selection bias is peaked in two cases depending on the value of $\alpha$. First,

when $\eta$ and $\epsilon$ have the same extreme values (1 or $-1$) and $\alpha = 1$. This leads to maximal negative bias. Second, when $\eta$ and $\epsilon$ have extreme but different sign values (1 or $-1$) and $\alpha = -1$. This corresponds to maximal positive bias. Intuitively, conditioning on the collider variable $W$ introduces a spurious effect between the two causes $A$ on $Y$: any information "explaining away" one cause will make the other cause more plausible. Using the job hiring example (Figure 2), if there is maximum negative discrimination based on the political beliefs of the candidates ($\alpha = -1$) and we measure discrimination using only labor union records, while political belief and job hiring have strong but opposite effects on labor union membership, the selection bias will be maximum to the point it cancels out all positive discrimination and leads to a conclusion of no discrimination. Figure 9(b) shows also that selection bias disappears when $\epsilon$ is zero, but not when $\eta$ is zero. When $\epsilon \neq 0$, selection bias can be zero depending on the value of $\epsilon$ as follows: $\epsilon = 1$ and $\alpha = -\eta$ or $\epsilon = -1$ and $\alpha = \eta$. Overall, selection bias has linear relation with both $\alpha$ and $\eta$, whereas non-linear relation with $\epsilon$[5].

Similarly to confounding and selection, measurement bias (Figure 9(c)) is peaked when $\beta$ and $\gamma$ have extreme values (1.0 or $-1.0$) but when $\delta = 0$. This is expected as, by definition, the more $Z$ and $T$ are independent, the higher measurement bias is. Conversely, the plot shows that measurement bias fades away as $\delta$ departs from 0[6].

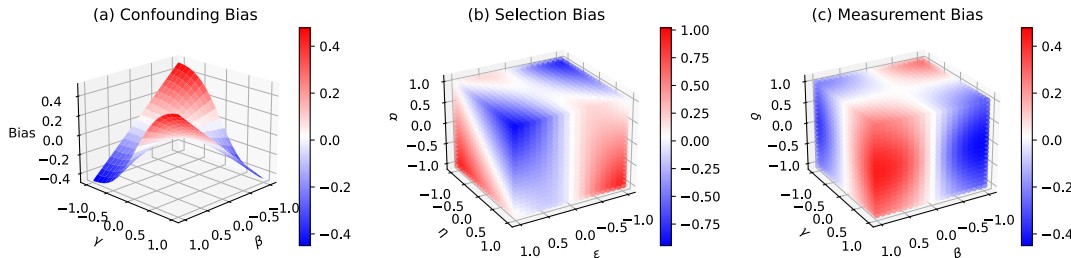

Figure 9: Bias magnitude in the linear case

## 8 BIAS MAGNITUDE: AN EMPIRICAL ANALYSIS

We use well-known fairness benchmark data sets Le Quy et al. (2022) for the experiments on real data: Adult [7], Boston housing [8], Compas Angwin et al. (2022), Communities and crimes [9] and Dutch census [10] data. The causal experiments on the real data are limited by the availability of true causal graphs for the benchmark fairness datasets. Furthemore, Binkytė-Sadauskienė et al. (2022) shows, that obtaining reliable causal graphs with causal discovery algorithms is a complicated task. However, we assume that the graphs in the literature are true for a given real dataset. We use the graphs by Zhang et al. (2018); Huan et al. (2020) for Adult and Dutch data sets to measure the interaction bias. For measuring confounder and collider biases we rely on graphs obtained by Binkytė-Sadauskienė et al. (2022) for Communities and Crimes, Boston Housing, Compas, and Dutch datasets (Appendix C). We estimate the measurement bias in the synthetic data(Appendix D.1). because the required structure is not present in the available graphs for the benchmark data sets. Although we cannot claim that the causal structure that we use for the experiments is the ground truth, it is useful for experimentally demonstrating the behavior of causal biases. In addition, the considered causal structures mots often show the presence of multiple causal biases at once. However, for the purposes of illustration, we control for a single type of bias separately. More precisely, we consider the difference in measured discrimination with the presence of the absence of a certain type of bias.

The experimental results for confounder bias show that the biases for each individual confounding variable are not significant (Figure 10). However, its magnitude increases and can erase the value

---

[5]Such relations can be observed more clearly using 2D plots. Figures 18 and Figure 19 in Appendix C
[6]The 2D plots in the appendix show clearly these observations.
[7]https://archive.ics.uci.edu/dataset/2/adult
[8]http://lib.stat.cmu.edu/datasets/boston
[9]https://archive.ics.uci.edu/dataset/183/communities+and+crime
[10]https://microdata.worldbank.org/index.php/catalog/2102/data-dictionary

for statistical disparity (Dutch data set), when multiple confounders are considered simultaneously (Figure 11). Measurement bias takes the highest value for *Synthetic2* dataset (Figure 13). The effect of $A$ on $Y$ when controlling for $T$ appears smaller than when controlling for $Z$. Here, the value of $T$ is highly dependent on $Z$ if $Z = 0$, but only loosely dependent on $Z$ if $Z = 1$. The prior probability of $Z$ conditions it to take value $Z = 1$ with probability $0.95$. Therefore, the link between $Z$ and $T$ is weak. The weak link between the variables makes $T$ a bad predictor for $Z$ and introduces a high measurement bias. Collider bias (Figure 12) is significant if it was introduced by conditioning on income (adult data), age (Compas data), economic status (Dutch data), poverty, unemployment, or divorce (Communities and crime data). Collider bias would reverse the value of statistical disparity, showing discrimination against the privileged group instead of discrimination against the disadvantaged group. We observe a portion of the interaction in all cases of the intersectional sensitive attribute (Figure 14). However, the value of synergism is negative, which means that it is not present in the data. Measurement of interaction bias for $A$ and $B$ individually can yield different values of interaction bias (Figure 15). Although the interaction term is symmetric for $A$ and $B$, the interaction bias value is also dependent on the probability $B = 1$ (when measuring *IntBias*(Y, A)) or $A = 1$ (when measuring *IntBias*(Y, B)). Therefore, for example, the interaction bias for sex is higher than for age in the Adult data set, because the probability of value 1 for age is higher than the probability of the sex variable taking value 1. Furthermore, we observe that the statistical disparity does not always correspond to the sum of interaction bias and statistical disparity without interaction $(StatDisp(Y, A) \neq SD_{Int}(Y, A) + P(b_1)Interaction(A, B))$, as required by equation 15. This observation suggests that the two sensitive variables $A$ and $B$ are not independent as suggested by the graphs provided by Zhang et al. (2018); Huan et al. (2020). Indeed, the graphs discovered by Binkytė-Sadauskienė et al. (2022) show the dependency between age and sex variables in the Dutch data set (Appendix C, Figure 22).

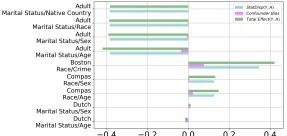

Figure 10: Confounder bias, when treating each confounder separately.

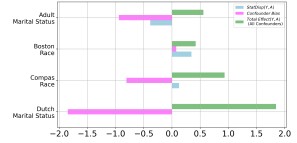

Figure 11: Confounder bias when treating all confounders together.

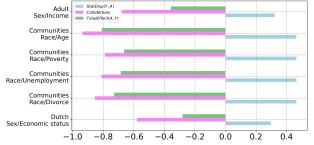

Figure 12: Collider bias, when treating each confounder separately.

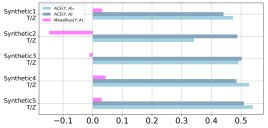

Figure 13: Measurement bias. Synthetic data.

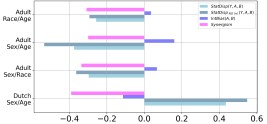

Figure 14: Interaction bias, intersectional sensitive variable.

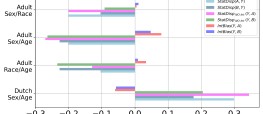

Figure 15: Interaction bias for individual sensitive attributes.

## 9    CONCLUSION

Several sources of bias have been described in the literature of Oxford (2021); Mehrabi et al. (2021). However, unlike existing work which typically do not define sources of bias formally, we provide closed-form expressions of a specific class of biases, namely causal biases. By analyzing the magnitude of bias in terms of the model parameters, we could establish an intuitive interpretation of bias based on the causal graph structure underlying each type of bias. Additionally, we provide in Appendix 9 an analysis of cases where two or more types of biases are present simultaneously. We strongly believe that a better understanding of the magnitude of causal biases, and more generally all sources of bias, will help ML fairness practitioners accurately predict the impact of proposed policies (e.g. training programs, awarness campaigns, establishing quotas, etc.) on existing discrimination.

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
