## A    PRELIMINARIES AND PREVIOUS RESULTS USED IN THE PROOFS

Variables are denoted by capital letters. In particular, $A$ is used for the sensitive variable (e.g., gender, race, age) and $Y$ is used for the outcome of the automated decision system (e.g., hiring, admission, releasing on parole). Small letters denote specific values of variables (e.g., $A = a'$, $W = w$). Bold capital and small letters denote a set of variables and a set of values, respectively.

Consider a pair of variables $X$ and $Y$. The variance of a variable $X$, $\sigma_x{}^2$, is a measure of dispersion which quantifies how far a set of values deviate from their mean and is defined as: $\sigma_x{}^2 = \mathbb{E}[X - \mathbb{E}[X]]^2$. Covariance of $X$ and $Y$, $\sigma_{xy}$, is a measure of the joint variability of two random variables and is defined as: $\sigma_{xy} = \mathbb{E}[[X - \mathbb{E}[X]][Y - \mathbb{E}[Y]]]$. Assuming a linear relationship between $X$ and $Y$ ($X$ is the predictor variable, while $Y$ is the response variable), the regression coefficient of $Y$ given $X$, $\beta_{yx}$, represents the slope of the regression line in the prediction of $Y$ given $X$ ($\frac{\partial}{\partial x}\mathbb{E}[Y|X = x]$) and is equal to $\beta_{yx} = \frac{\sigma_{xy}}{\sigma_x{}^2}$. Correlation coefficient $\rho_{yx}$, however, represents the slope of the least square error line in the prediction of $Y$ given $X$. The relationships between $\sigma_{yx}$, $\beta_{yx}$, and $\rho_{yx}$ are as follows:

$$\beta_{yx} = \frac{\sigma_{yx}}{\sigma_x^2} = \rho_{yx}\frac{\sigma_y}{\sigma_x}$$

$$\rho_{yx} = \rho_{xy} = \frac{\sigma_{yx}}{\sigma_x\sigma_y} = \beta_{yx}\frac{\sigma_x}{\sigma_y} = \beta_{xy}\frac{\sigma_y}{\sigma_x}$$

Partial regression coefficient, $\beta_{yx.z}$, represents the slope of the regression line of $Y$ on $X$ when we hold variable $Z$ constant ($\frac{\partial}{\partial x}\mathbb{E}[Y|X = x, Z = z]$). A well known result by Cramer Cramér (1999) allows to express $\beta_{yx.z}$ in terms of covariance between pairs of variables Pearl (2013):

$$\beta_{yx.z} = \frac{\sigma_z{}^2\sigma_{xy} - \sigma_{yz}\sigma_{zx}}{\sigma_x{}^2\sigma_z{}^2 - \sigma_{xz}{}^2} \tag{20}$$

For standardized variables (all variables are normalized to have a zero mean and a unit variance), the partial regression coefficient has a simpler expression since $\beta_{yx} = \sigma_{yx}$:

$$\beta_{yx.z} = \frac{\sigma_{xy} - \sigma_{yz}\sigma_{zx}}{1 - \sigma_{xz}{}^2} \tag{21}$$

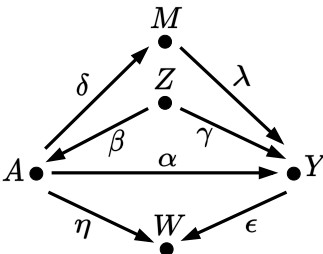

Figure 16: Causal graph with linearly related variables. Arrow labels represent linear regression coefficients.

Another known result by Wright Wright (1921); Pearl (2013) allows to represent the covariance of two variables in terms of the regression coefficients of the different paths (causal and non-causal, but not passing through any collider variable) between those two variables. More precisely, $\sigma_{yx}$ is equal to the sum of the regression coefficients of every path between $x$ and $y$, weighted by the variance of the root variable of each path. For instance, in Figure 16, $\sigma_{ya} = \sigma_a{}^2\alpha + \sigma_z{}^2\beta\gamma + \sigma_a{}^2\delta\lambda$. Notice that the coefficients $\eta$ and $\epsilon$ are not included as the path $A \to W \leftarrow Y$ is not $d-$connected ($W$ is a collider variable). For standardized variables, the expression is simpler as all variables are normalized to have a unit variance. For the same example (Figure 16), $\sigma_{ya} = \alpha + \beta\gamma + \delta\lambda$. For linear models, regression coefficients can be interpreted causally. For instance, using the same example of Figure 16, $\alpha$ repesents the direct causal effect of $A$ on $Y$. In more general models, the causal effect

between two variables is typically expressed in terms of intervention probabilities. Intervention, noted $do(V = v)$ Pearl (2009), is a manipulation of the model that consists in fixing the value of a variable (or a set of variables) to a specific value regardless of the causes of that variable. The intervention $do(V = v)$ induces a different distribution on the other variables. Intuitively, while $\mathbb{P}(Y|A = a)$ reflects the population distribution of $Y$ among individuals whose $A$ value is $a$, $\mathbb{P}(Y|do(A = a)$[11] reflects the population distribution of $Y$ if *everyone in the population* had their $A$ value fixed at $a$. The obtained distribution $\mathbb{P}(Y|do(A = a)$ can be considered as a *counterfactual* distribution since the intervention forces $a$ to take a value different from the one it would take in the actual world. $\mathbb{P}(Y|do(A = a)$ is not always computable from the data, a problem known as identifiability. For instance, if all counfounder variables are observable, the intervention probability, $\mathbb{P}(Y|do(A = a))$, can be computed by adjusting on the counfounder(s). For instance, assuming $Z$ is the only confounder of $A$ and $Y$,

$$\mathbb{P}(Y|do(A = a)) = \sum_{z \in Z} \mathbb{P}(Y|A = a, Z = z).\mathbb{P}(Z = z) \tag{22}$$

Equation 22 is called the backdoor formula.

## A.1 STATISTICAL DISPARITY

Statistical disparity Rawls (2020) between groups $A = 0$ and $A = 1$, denoted as $statDisp(Y, A)$, is the difference between the conditional probabilities: $\mathbb{P}(y_1|a_1) - \mathbb{P}(y_1|a_0)$:

**Definition A.1.**

$$StatDisp(Y, A) = \mathbb{P}(y_1|a_1) - \mathbb{P}(y_1|a_0). \tag{23}$$

In presence of a confounder variable, $Z$, between $A$ and $Y$, statistical disparity is a biased estimation of the discrimination as it does not filter out the spurious effect due to the confounding. For the sake of the proofs, we define the following variant of statistical disparity:

**Definition A.2.**

$$StatDisp(Y, A)_Z = \sum_{z \in Z} \big(\mathbb{P}(y_1|a_1, z) - \mathbb{P}(y_1|a_0, z)\big).\mathbb{P}(z). \tag{24}$$

Notice that if $Z$ d-separates $A$ and $Y$, $StatDisp(Y, A)_Z$ coincides with the average causal effect $ACE$ which defined using the do-operator (Equation 22):

**Definition A.3.**

$$ACE(Y, A) = \mathbb{P}(y_1|do(a_1)) - \mathbb{P}(y_1|do(a_0)). \tag{25}$$

## B PROOFS

### B.1 PROOF OF THEOREM 3.1

**Definition B.1.** *Confounding bias is defined as*[12]:

$$ConfBias(Y, A) = StatDisp(Y, A) - ACE(Y, A) \tag{26}$$

*Proof.* Let $\mathbb{P}(z_1) = \epsilon$ ($\epsilon \in ]0, 1[$) and hence $\mathbb{P}(z_0) = 1 - \epsilon$. And let $\mathbb{P}(y_1|a_0, z_0) = \alpha$, $\mathbb{P}(y_1|a_0, z_1) = \beta$, $\mathbb{P}(y_1|a_1, z_0) = \gamma$, and $\mathbb{P}(y_1|a_1, z_1) = \delta$. Finally, let $\mathbb{P}(z_0|a_0) = \tau$. The remaining conditional probabilities of $Z$ given $A$ are equal to the following:

$$\mathbb{P}(z_1|a_0) = 1 - \mathbb{P}(z_0|a_0) = 1 - \tau \tag{27}$$

$$\mathbb{P}(z_1|a_1) = \frac{\mathbb{P}(z_1) - \mathbb{P}(z_1|a_0)\mathbb{P}(a_0)}{\mathbb{P}(a_1)}$$

$$= 2\epsilon + \tau - 1 \tag{28}$$

$$\mathbb{P}(z_0|a_1) = 1 - \mathbb{P}(z_1|a_1) \tag{29}$$

$$= 2 - 2\epsilon - \tau$$

---

[11]The notations $Y_{A \leftarrow a}$ and $Y(a)$ are used in the literature as well. $\mathbb{P}(Y = y|do(A = a)) = \mathbb{P}(Y_{A=a} = y) = \mathbb{P}(Y_a = y) = \mathbb{P}(y_a)$ is used to define the causal effect of $A$ on $Y$.

[12]In this paper, bias is defined by substracting the correct value of discrimination from the biased estimation.

Equations (27) and (29) follow from the fact that, given $u_i$ events are exhaustive and mutually exclusive, $\sum_i \mathbb{P}(a_i|X) = 1$. Equation (28) follows from the fact that, given $u_i$ events are exhaustive and mutually exclusive, $\sum_i \mathbb{P}(X|u_i)\mathbb{P}(u_i) = \mathbb{P}(X)$. $StatDisp(Y, A)$ can then be expressed in terms of the above parameters:

$$
\begin{aligned}
\mathbb{P}(y_1|a_1) - \mathbb{P}(y_1|a_0) &= \sum_{z \in Z} (\mathbb{P}(y_1|a_1, z)\mathbb{P}(z|a_1) - \mathbb{P}(y_1|a_0, z)\mathbb{P}(z|a_0) \\
&= \mathbb{P}(y_1|a_1, z_0)\mathbb{P}(z_0|a_1) - \mathbb{P}(y_1|a_0, z_0)\mathbb{P}(z_0|a_0) \\
&\quad + \mathbb{P}(y_1|a_1, z_1)\mathbb{P}(z_1|a_1) - \mathbb{P}(y_1|a_0, z_1)\mathbb{P}(z_1|a_0) \\
&= \gamma(2 - 2\epsilon - \tau) - \alpha\tau + \delta(2\epsilon + \tau - 1) - \beta(1 - \tau) \\
&= \tau(-\alpha + \beta - \gamma + \delta) + 2\epsilon(\delta - \gamma) + 2\gamma - \delta - \beta
\end{aligned}
$$

$ACE(Y, A)$, on the other hand can be expressed as follows:

$$
\begin{aligned}
\mathbb{P}(y_1|do(a_1)) - \mathbb{P}(y_1|do(a_0)) &= \sum_{z \in Z} (\mathbb{P}(y_1|a_1, z) - \mathbb{P}(y_1|a_0, z))\mathbb{P}(z) \\
&= \mathbb{P}(y_1|a_1, z_0) - \mathbb{P}(y_1|a_0, z_0))\mathbb{P}(z_0) \\
&\quad + \mathbb{P}(y_1|a_1, z_1) - \mathbb{P}(y_1|a_0, z_1))\mathbb{P}(z_1) \\
&= (\gamma - \alpha)(1 - \epsilon) + (\delta - \beta)\epsilon
\end{aligned}
$$

Confounding bias is then equal to:

$$
\begin{aligned}
StatDisp(Y, A) - ACE(Y, A) &= \mathbb{P}(y_1|a_1) - \mathbb{P}(y_1|a_0) - (\mathbb{P}(y_1|do(a_1)) - \mathbb{P}(y_1|do(a_0))) \\
&= \tau(-\alpha + \beta - \gamma + \delta) + 2\epsilon(\delta - \gamma) + 2\gamma - \delta - \beta \\
&\quad - ((\gamma - \alpha)(1 - \epsilon) + (\delta - \beta)\epsilon) \\
&= \tau(-\alpha + \beta - \gamma + \delta) + 2\epsilon\delta - 2\epsilon\gamma + 2\gamma - \delta - \beta \\
&\quad - \gamma + \gamma\epsilon + \alpha - \alpha\epsilon - \delta\epsilon + \beta\epsilon \\
&= \tau(-\alpha + \beta - \gamma + \delta) + \epsilon(2\delta - 2\gamma + \gamma - \alpha - \delta + \beta) \\
&\quad + 2\gamma - \delta - \beta - \gamma + \alpha \\
&= \tau(-\alpha + \beta - \gamma + \delta) + \epsilon(-\alpha + \beta - \gamma + \delta) + \alpha - \beta + \gamma - \delta \\
&= (1 - \tau - \epsilon)(\alpha - \beta + \gamma - \delta)
\end{aligned}
$$

$\square$

**Theorem B.2.** *Assuming $A$, $Y$, and $Z$ binary variables, the difference in discrimination due to confounding is equal to:*

$$
\begin{aligned}
ConfBias(Y, A) &= (1 - \mathbb{P}(z_0|a_0) - \mathbb{P}(z_1)) \\
&\quad \times (\alpha - \beta - \gamma + \delta + \frac{\gamma}{\mathbb{P}(a_1)} - \frac{\delta}{\mathbb{P}(a_1)}).
\end{aligned}
\tag{30}
$$

*where, $\alpha = \mathbb{P}(y_1|a_0, z_0)$, $\beta = \mathbb{P}(y_1|a_0, z_1)$, $\gamma = \mathbb{P}(y_1|a_1, z_0)$, and $\delta = \mathbb{P}(y_1|a_1, z_1)$.*

### B.2 PROOF OF THEOREM 3.2

*Proof.* For Equation (2),

$$
\begin{aligned}
ConfBias(Y, A) &= \beta_{ya} - \beta_{ya.z} \\
&= \frac{\sigma_{ya}}{\sigma_a^2} - \frac{\sigma_z^2 \sigma_{ya} - \sigma_{yz}\sigma_{za}}{\sigma_a^2 \sigma_z^2 - \sigma_{za}^2} \\
&= \frac{\frac{\sigma_{ya}}{\sigma_a^2}(\sigma_a^2 \sigma_z^2 - \sigma_{za}^2) - (\sigma_z^2 \sigma_{ya} - \sigma_{yz}\sigma_{za})}{\sigma_a^2 \sigma_z^2 - \sigma_{za}^2} \\
&= \frac{\frac{\sigma_{ya}}{\sigma_a^2}\sigma_a^2 \sigma_z^2 - \frac{\sigma_{ya}}{\sigma_a^2}\sigma_{za}^2 - \sigma_z^2 \sigma_{ya} + \sigma_{yz}\sigma_{za}}{\sigma_a^2 \sigma_z^2 - \sigma_{za}^2} \\
&= \frac{\sigma_{za}\sigma_{yz} - \frac{\sigma_{ya}}{\sigma_a^2}\sigma_{za}^2}{\sigma_a^2 \sigma_z^2 - \sigma_{za}^2}
\end{aligned}
$$

For Equation (3),

$$ConfBias(Y, A) = \beta_{ya} - \beta_{ya.z}$$

$$= \frac{\sigma_{ya}}{\sigma_a^2} - \frac{\sigma_z^2 \sigma_{ya} - \sigma_{yz}\sigma_{za}}{\sigma_a^2 \sigma_z^2 - \sigma_{za}^2}$$

$$= \frac{\sigma_a^2 \alpha + \sigma_z^2 \beta\gamma}{\sigma_a^2} - \frac{\sigma_z^2(\sigma_a^2 \alpha + \sigma_z^2 \beta\gamma) - (\sigma_z^2\gamma + \sigma_z^2\beta\alpha)(\sigma_z^2\beta)}{\sigma_a^2 \sigma_z^2 - (\sigma_z^2\beta)^2}$$

$$= \frac{\cancel{\sigma_a^2}\alpha}{\cancel{\sigma_a^2}} + \frac{\sigma_z^2 \beta\gamma}{\sigma_a^2} - \frac{\sigma_z^2\sigma_a^2\alpha + \cancel{\sigma_z^4\beta\gamma} - \cancel{\sigma_z^4\beta\gamma} - \sigma_z^4\beta^2\alpha}{\sigma_a^2\sigma_z^2 - \sigma_z^4\beta^2}$$

$$= \cancel{\alpha} + \frac{\sigma_z^2 \beta\gamma}{\sigma_a^2} - \frac{\cancel{\alpha}\cancel{(\sigma_z^2\sigma_a^2 - \sigma_z^4\beta^2)}}{\cancel{\sigma_a^2\sigma_z^2 - \sigma_z^4\beta^2}}$$

$$= \frac{\sigma_z^2}{\sigma_a^2}\beta\gamma$$

$\square$

## B.3 PROOF OF THEOREM B.3

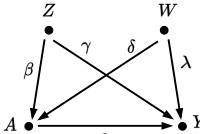

Figure 17: Confounding structure with two confounders

**Theorem B.3.** *Let $A, Y, Z, W$ variables as in Figure 17. Assuming that all variables are standardized and that $W$ and $Z$ are independent, the regression coefficent of $Y$ on $A$ conditioning on $Z$ and $W$, the confounding bias is equal:*

$$ConfBias(Y, A) = \frac{\sigma_{za}\sigma_{yz} + \sigma_{wa}\sigma_{yw} - \sigma_{ya}(\sigma_{za}^2 + \sigma_{wa}^2)}{1 - \sigma_{za}^2 - \sigma_{wa}^2} \tag{31}$$

*And in terms of the regression coefficients:*

$$ConfBias(Y, A) = \beta\gamma + \delta\lambda \tag{32}$$

It is important to mention that although Theorem B.3 assumes that the variables are standardized, the equations can be easily generalized to the non-standardized variables case. Moreover, the proof is general and can be extended to the case where the two confounders are not independent.

*Proof.* The proof is based on proving that:

$$\beta_{ya.zw} = \frac{\sigma_{ya} - \sigma_{za}\sigma_{yz} - \sigma_{wy}\sigma_{wa}}{1 - \sigma_{za}^2 - \sigma_{wa}^2} \tag{33}$$

From Cramér Cramér (1999) (Page 307), we know that the partial regression coefficient can be expressed as:

$$\beta_{ya.zw} = \rho_{ya.zw}\frac{\sigma_{y.zw}}{\sigma_{a.zw}} \tag{34}$$

Where $\rho_{ya.zw}$ denotes the partial correlation and $\sigma_{a.zw}, \sigma_{y.zw}$ denote the residual variances.

Based on the correlation matrix:

$$\begin{bmatrix} 1 & \rho_{ya} & \rho_{yz} & \rho_{yw} \\ \rho_{ay} & 1 & \rho_{az} & \rho_{aw} \\ \rho_{zy} & \rho_{za} & 1 & \rho_{zw} \\ \rho_{wy} & \rho_{wa} & \rho_{wz} & 1 \end{bmatrix}$$

the partial correlation $\rho_{ya.zw}$ can be expressed in terms of cofactors as follows[13]:

$$\rho_{ya.zw} = -\frac{C_{ya}}{\sqrt{C_{yy}C_{aa}}} \tag{35}$$

where $C_{ij}$ denotes the cofactor of the element $\rho_{ij}$ in the determinant of the correlation matrix and are equal to the following:

$$\begin{aligned} C_{ya} = -(&\rho_{ya} - \rho_{ya}\rho_{zw}^2 - \rho_{za}\rho_{yz} - \rho_{wa}\rho_{yw} \\ &+ \rho_{za}\rho_{yw}\rho_{wz} + \rho_{wa}\rho_{yz}\rho_{zw}) \end{aligned} \tag{36}$$

$$C_{yy} = 1 - \rho_{zw}^2 - \rho_{za}^2 - \rho_{wa}^2 + 2\rho_{za}\rho_{aw}\rho_{wz} \tag{37}$$

$$C_{aa} = 1 - \rho_{zw}^2 - \rho_{zy}^2 - \rho_{wy}^2 + 2\rho_{yz}\rho_{yw}\rho_{wz} \tag{38}$$

Residual variances in Equation 34 can be expressed in terms of total and partial correlation coefficients as follows Cramér (1999)(Equation 23.4.5 in page 307):

$$\sigma_{y.zw}^2 = \sigma_y^2(1 - \rho_{yz}^2)(1 - \rho_{yw.z}^2)(1 - \rho_{ya.zw}^2) \tag{39}$$

$$\sigma_{a.zw}^2 = \sigma_a^2(1 - \rho_{az}^2)(1 - \rho_{aw.z}^2)(1 - \rho_{ay.zw}^2) \tag{40}$$

As the last term is the same, we have:

$$\frac{\sigma_{y.zw}}{\sigma_{a.zw}} = \frac{\sigma_y\sqrt{(1 - \rho_{yz}^2)(1 - \rho_{yw.z}^2)}}{\sigma_a\sqrt{(1 - \rho_{az}^2)(1 - \rho_{aw.z}^2)}} \tag{41}$$

The partial correlation coefficients in Equation 41 can be expressed in terms of total correlation coefficients as follows Cramér (1999) (Equation 23.4.3 in page 306):

$$\rho_{yw.z} = \frac{\rho_{yw} - \rho_{yz}\rho_{wz}}{\sqrt{(1 - \rho_{yz}^2)(1 - \rho_{wz}^2)}} \tag{42}$$

After simple algebraic steps, we obtain:

$$\frac{\sigma_{y.zw}}{\sigma_{a.zw}} = \frac{\sigma_y}{\sigma_a}\frac{\sqrt{1 - \rho_{zw}^2 - \rho_{yz}^2 - \rho_{yw}^2 + 2\rho_{zy}\rho_{yw}\rho_{wz}}}{\sqrt{1 - \rho_{zw}^2 - \rho_{az}^2 - \rho_{aw}^2 + 2\rho_{za}\rho_{aw}\rho_{wz}}} \tag{43}$$

Finally, $\beta_{ya.zw}$ in Equation 34 can be expressed in terms of total correlation coefficients as follows:

$$\beta_{ya.zw} = \frac{\sigma_y}{\sigma_a}\frac{Q}{1 - \rho_{zw}^2 - \rho_{za}^2 - \rho_{wa}^2 + 2\rho_{za}\rho_{aw}\rho_{wz}} \tag{44}$$

where

$$\begin{aligned} Q = &\rho_{ya} - \rho_{ya}\rho_{zw}^2 - \rho_{za}\rho_{yz} - \rho_{wy}\rho_{wa} \\ &+ \rho_{za}\rho_{yw}\rho_{zw} + \rho_{wa}\rho_{yz}\rho_{zw} \end{aligned}$$

Recall that $\rho_{ya} = \frac{\sigma_{ya}}{\sigma_y\sigma_a}$. The formula becomes:

$$\beta_{ya.zw} = \frac{Q}{R} \tag{45}$$

Where

$$\begin{aligned} Q = &\sigma_{ya}(\sigma_z^2\sigma_w^2 - \sigma_{zw}^2) + \sigma_{yz}(\sigma_{wa}\sigma_{zw} - \sigma_{za}\sigma_w^2) \\ &+ \sigma_{wy}(\sigma_{za}\sigma_{zw} - \sigma_{wa}\sigma_z^2) \end{aligned}$$

---

[13]The proof is sketched in https://en.wikipedia.org/wiki/Partial_correlation.

And

$$R = \sigma_a^2 \sigma_z^2 \sigma_w^2 - \sigma_a^2 \sigma_{zw}^2 - \sigma_a \sigma_z \sigma_w^2 \sigma_{za}^2$$
$$- \sigma_z^2 \sigma_{aw}^2 + 2\sigma_a \sigma_z \sigma_{az} \sigma_{aw} \sigma_{zw}$$

For standardized variables, $\forall v, \sigma_v = 1$, and hence $\forall u, v\sigma_{uv} = \rho_{uv}$. Equation 44 becomes:

$$\beta_{ya.zw} = \frac{Q}{1 - \sigma_{zw}^2 - \sigma_{za}^2 - \sigma_{wa}^2 + 2\sigma_{za}\sigma_{aw}\sigma_{wz}} \tag{46}$$

Where

$$Q = \sigma_{ya}(1 - \sigma_{zw}^2) + \sigma_{yz}(\sigma_{wa}\sigma_{zw} - \sigma_{za})$$
$$+ \sigma_{yw}(\sigma_{za}\sigma_{zw} - \sigma_{wa})$$

If we further assume that confounders are uncorrelated, that is, $\sigma_{zw} = 0$, then we have the simpler expression:

$$\beta_{ya.zw} = \frac{\sigma_{ya} - \sigma_{za}\sigma_{yz} - \sigma_{wy}\sigma_{wa}}{1 - \sigma_{za}^2 - \sigma_{wa}^2} \tag{47}$$

For Equation (31):

$$\begin{aligned}
ConfBias(Y, A) &= \beta_{ya} - \beta_{ya.zw} \\
&= \sigma_{ya} - \frac{\sigma_{ya} - \sigma_{za}\sigma_{yz} - \sigma_{wa}\sigma_{yw}}{1 - \sigma_{za}^2 - \sigma_{wa}^2} \\
&= \frac{\sigma_{ya}(1 - \sigma_{za}^2 - \sigma_{wa}^2) - \sigma_{ya} + \sigma_{za}\sigma_{yz} + \sigma_{wa}\sigma_{yw}}{1 - \sigma_{za}^2 - \sigma_{wa}^2} \\
&= \frac{\cancel{\sigma_{ya}} - \sigma_{ya}\sigma_{za}^2 - \sigma_{ya}\sigma_{wa}^2 - \cancel{\sigma_{ya}} + \sigma_{za}\sigma_{yz} + \sigma_{wa}\sigma_{yw}}{1 - \sigma_{za}^2 - \sigma_{wa}^2} \\
&= \frac{\sigma_{za}\sigma_{yz} + \sigma_{wa}\sigma_{yw} - \sigma_{ya}(\sigma_{za}^2 + \sigma_{wa}^2)}{1 - \sigma_{za}^2 - \sigma_{wa}^2}
\end{aligned} \tag{48}$$

For Equation (32):

$$\begin{aligned}
ConfBias(Y, A) &= \beta_{ya} - \beta_{ya.zw} \\
&= \alpha + \beta\gamma + \lambda\delta - \frac{\alpha + \beta\gamma + \lambda\delta - \beta(\gamma + \beta\alpha) - \delta(\lambda + \delta\alpha)}{1 - \beta^2 - \delta^2} \\
&= \alpha + \beta\gamma + \lambda\delta - \frac{\alpha + \cancel{\beta\gamma} + \cancel{\lambda\delta} - \cancel{\beta\gamma} - \beta^2\alpha - \cancel{\delta\lambda} - \delta^2\alpha}{1 - \beta^2 - \delta^2} \\
&= \cancel{\alpha} + \beta\gamma + \lambda\delta - \frac{\cancel{\alpha(1 - \beta^2 - \delta^2)}}{\cancel{1 - \beta^2 - \delta^2}} \\
&= \beta\gamma + \lambda\delta
\end{aligned} \tag{49}$$

$\square$

## B.4  PROOF OF THEOREM 4.1

**Definition B.4.** *Given the basic collider structure (Figure 6), selection bias is defined as:*

$$SelBias(Y, A, W) = StatDisp(Y, A)_W - StatDisp(Y, A) \tag{50}$$

*Proof.* The proof is based on the proof of Theorem 3.1. Notice that, conditioning on variable $Z$ in $ACE(Y, A)$ has the same formulation as conditioning on $W$ in $StatDisp(Y, A)_W$. The difference

is that the conditioning is on $W$ instead of $Z$. The other important difference is that in Theorem 3.1, the unconditional expression $StatDisp(A, Y)$ is the biased estimation of the discrimination and the conditional expression $ACE(Y, A)$ is the unbiased estimation. Whereas in Theorem 4.1, it is the opposite: the unconditional expression $StatDisp(A, Y)$ is the unbiased estimation of discrimination and the conditional expression $StatDisp(Y, A)_W$ is the biased estimation. Hence, selection bias is just the opposite of Equation (1) while replacing the variable $Z$ by the variable $W$. □

## B.5 Proof of Theorem 4.2

*Proof.* For Equation (7),

$$SelBias(Y, A) = \beta_{ya.w} - \beta_{ya}$$

$$= \frac{\sigma_w^2 \sigma_{ya} - \sigma_{yw}\sigma_{wa}}{\sigma_a^2 \sigma_w^2 - \sigma_{wa}^2} - \frac{\sigma_{ya}}{\sigma_a^2}$$

$$= \frac{(\sigma_w^2 \sigma_{ya} - \sigma_{yw}\sigma_{wa}) - \frac{\sigma_{ya}}{\sigma_a^2}(\sigma_a^2 \sigma_w^2 - \sigma_{wa}^2)}{\sigma_a^2 \sigma_w^2 - \sigma_{wa}^2}$$

$$= \frac{\sigma_w^2 \sigma_{ya} - \sigma_{yw}\sigma_{wa} - \frac{\sigma_{ya}}{\sigma_a^2}\sigma_a^2\sigma_w^2 + \frac{\sigma_{ya}}{\sigma_a^2}\sigma_{wa}^2}{\sigma_a^2 \sigma_w^2 - \sigma_{wa}^2}$$

$$= \frac{\frac{\sigma_{ya}}{\sigma_a^2}\sigma_{wa}^2 - \sigma_{wa}\sigma_{yw}}{\sigma_a^2 \sigma_w^2 - \sigma_{wa}^2}$$

For Equation (8),

$$SelBias(Y, A) = \beta_{ya.w} - \beta_{ya}$$

$$= \frac{\sigma_w^2 \sigma_{ya} - \sigma_{yw}\sigma_{wa}}{\sigma_a^2 \sigma_w^2 - \sigma_{wa}^2} - \frac{\sigma_{ya}}{\sigma_a^2}$$

$$= \frac{\sigma_w^2 \sigma_a^2 \alpha - (\sigma_y^2 \epsilon + \sigma_a^2 \alpha\eta)(\sigma_a^2 \eta + \sigma_a^2 \alpha\epsilon)}{\sigma_a^2 \sigma_w^2 - (\sigma_a^2 \eta + \sigma_a^2 \alpha\epsilon)^2} - \frac{\sigma_a^2 \alpha}{\sigma_a^2}$$

$$= \frac{\sigma_w^2 \sigma_a^2 \alpha - \sigma_y^2 \sigma_a^2 \epsilon\eta - \sigma_y^2 \sigma_a^2 \alpha\epsilon^2 - \sigma_a^4 \alpha\eta^2 - \sigma_a^4 \alpha^2 \eta\epsilon}{\sigma_a^2 \sigma_w^2 - (\sigma_a^2 \eta + \sigma_a^2 \alpha\epsilon)^2}$$

$$\quad - \frac{\alpha(\sigma_a^2 \sigma_w^2 - (\sigma_a^2 \eta + \sigma_a^2 \alpha\epsilon)^2)}{\sigma_a^2 \sigma_w^2 - (\sigma_a^2 \eta + \sigma_a^2 \alpha\epsilon)^2}$$

$$= \frac{\sigma_w^2 \sigma_a^2 \alpha - \sigma_y^2 \sigma_a^2 \epsilon\eta - \sigma_y^2 \sigma_a^2 \alpha\epsilon^2 - \sigma_a^4 \alpha\eta^2 - \sigma_a^4 \alpha^2 \eta\epsilon}{\sigma_a^2 \sigma_w^2 - (\sigma_a^2 \eta + \sigma_a^2 \alpha\epsilon)^2}$$

$$\quad + \frac{-\sigma_a^2 \sigma_w^2 \alpha + \sigma_a^4 \alpha\eta^2 + 2\sigma_a^4 \alpha^2 \eta\epsilon + \sigma_a^4 \alpha^3 \epsilon^2}{\sigma_a^2 \sigma_w^2 - (\sigma_a^2 \eta + \sigma_a^2 \alpha\epsilon)^2}$$

$$= \epsilon \frac{\sigma_a^4 \alpha^2 \eta + \sigma_a^4 \alpha^3 \epsilon - \sigma_y^2 \sigma_a^2 \eta - \sigma_y^2 \sigma_a^2 \alpha\epsilon}{\sigma_a^2 \sigma_w^2 - (\sigma_a^2 \eta + \sigma_a^2 \alpha\epsilon)^2}$$

□

## B.6 Proof of Theorem 5.1

**Definition B.5.** *Given variables $A$, $Y$, $Z$, and $T$ with causal relations as in Figure 7, measurement bias can be defined as:*

$$MeasBias(Y, A) = StatDisp(Y, A)_T - StatDisp_Z(Y, A) \tag{51}$$

*Proof.* Let $\mathbb{P}(t_1) = \epsilon$ ($\epsilon \in ]0, 1[$) and hence $\mathbb{P}(t_0) = 1 - \epsilon$. And let $\mathbb{P}(y_1|a_0, t_0) = \alpha$, $\mathbb{P}(y_1|a_0, t_1) = \beta$, $\mathbb{P}(y_1|a_1, t_0) = \gamma$, and $\mathbb{P}(y_1|a_1, t_1) = \delta$. Finally, let $\mathbb{P}(t_0|a_0) = \tau$. The remaining conditional

probabilities of $T$ given $A$ are equal to the following:

$$\mathbb{P}(t_1|a_0) = 1 - \mathbb{P}(t_0|a_0) = 1 - \tau$$

$$\mathbb{P}(t_1|a_1) = \frac{\mathbb{P}(t_1) - \mathbb{P}(t_1|a_0)\mathbb{P}(a_0)}{\mathbb{P}(a_1)}$$

$$= 2\epsilon + \tau - 1 \tag{52}$$

$$\mathbb{P}(z_0|a_1) = 1 - \mathbb{P}(z_1|a_1) \tag{53}$$

$$= 2 - 2\epsilon - \tau$$

According to Definition B.5:

$$MeasBias(Y, A) = ACE(Y, A)_T - ACE(Y, A)$$

By the proof of Theorem 3.1, the first term:

$$ACE(Y, A)_T = \epsilon(\delta - \beta) + (1 - \epsilon)(\gamma - \alpha)$$

The rest of the proof consists in expressing $ACE(Y, A)$ in terms of the error term $\mathbb{P}(T|Z)$.

$$ACE(Y, A) = \mathbb{P}(y_1|do(a_1))_Z - \mathbb{P}(y_1|do(a_0))_Z \tag{54}$$

where:

$$\mathbb{P}(y_1|do(a))_Z =$$

$$\frac{\mathbb{P}(y_1, a, t_1)}{\mathbb{P}(a|t_1)} \frac{\left(1 - \frac{\mathbb{P}(t_1|z_0)}{\mathbb{P}(t_1|a,y_1)}\right)\left(1 - \frac{\mathbb{P}(t_1|z_0)}{\mathbb{P}(t_1)}\right)}{1 - \mathbb{P}(t_1|z_0)\frac{\mathbb{P}(a)}{\mathbb{P}(t_1)}}$$

$$+ \frac{\mathbb{P}(y_1, a, t_0)}{\mathbb{P}(a|t_0)} \frac{\left(1 - \frac{\mathbb{P}(t_0|z_1)}{\mathbb{P}(t_0|a,y_1)}\right)\left(1 - \frac{\mathbb{P}(t_0|z_1)}{\mathbb{P}(t_0)}\right)}{1 - \mathbb{P}(t_0|z_1)\frac{\mathbb{P}(a)}{\mathbb{P}(t_0)}} \tag{55}$$

The proof can be found in Pearl (2010) (Section 3). Using Bayes rule, we can easily show that

$$\mathbb{P}(y_1, a_1, t_1) = \epsilon\delta + \frac{\delta\tau}{2} - \frac{\delta}{2}$$

$$\mathbb{P}(y_1, a_1, t_0) = \gamma - \epsilon\gamma + \frac{\delta\tau}{2} - \frac{\tau\gamma}{2}$$

$$\mathbb{P}(y_1, a_0, t_1) = \frac{\beta}{2} - \frac{\beta\tau}{2}$$

$$\mathbb{P}(y_1, a_0, t_0) = \frac{\gamma\tau}{2}$$

Using Bayes rule and the marginal conditional probability rule: $\mathbb{P}(A|B) = \sum_{z\in Z}\mathbb{P}(A|B, z)\mathbb{P}(z|B)$, we can easily show that:

$$\mathbb{P}(t_1|a_0, y_1) = \frac{1}{4}\frac{\beta - \beta\tau}{\alpha\tau + \beta - \beta\tau}$$

$$\mathbb{P}(t_0|a_0, y_1) = \frac{1}{4}\frac{\gamma\tau}{\gamma\tau + \beta - \beta\tau}$$

$$\mathbb{P}(t_1|a_1, y_1) = \frac{\epsilon\delta + \frac{\delta\tau}{2} - \frac{\delta}{2}}{4\gamma - 4\epsilon\gamma - 2\tau\gamma + 4\epsilon\delta + 2\delta\tau - 2\delta}$$

$$\mathbb{P}(t_0|a_1, y_1) = \frac{\gamma - \epsilon\gamma - \frac{\tau\gamma}{2}}{4\gamma - 4\epsilon\gamma - 2\tau\gamma + 4\epsilon\delta + 2\delta\tau - 2\delta}$$

Finally, using Bayes rule, we can show that:

$$\mathbb{P}(a_0|t_1) = \frac{\mathbb{P}(t_1|a_0)\mathbb{P}(a_0)}{\mathbb{P}(t_1)} = \frac{(1 - \tau)}{2\epsilon}$$

$$\mathbb{P}(a_0|t_0) = \frac{\mathbb{P}(t_0|a_0)\mathbb{P}(a_0)}{\mathbb{P}(t_0)} = \frac{\tau}{2 - 2\epsilon}$$

$$\mathbb{P}(a_1|t_1) = \frac{\mathbb{P}(t_1|a_1)\mathbb{P}(a_1)}{\mathbb{P}(t_1)} = \frac{\epsilon + \frac{\tau}{2} - \frac{1}{2}}{\epsilon}$$

$$\mathbb{P}(a_1|t_0) = \frac{\mathbb{P}(t_0|a_1)\mathbb{P}(a_1)}{\mathbb{P}(t_0)} = \frac{(2 - 2\epsilon - \tau)}{2 - 2\epsilon}$$

After some algebra, we have:

$$
\begin{aligned}
ACE(Y, A) = \ & \epsilon\big(\delta - \beta + 4\mathbb{P}(t_1|z_0)(\beta - \delta + \gamma\Phi + \gamma\Psi)\big)\, Q \\
& + (1-\epsilon)\big(\gamma - \alpha + 4\mathbb{P}(t_0|z_1)(\alpha - \gamma + \delta + \delta\Psi^{-1} + \beta\Phi^{-1})\big)\, R
\end{aligned}
\tag{56}
$$

where

$$
\alpha = \mathbb{P}(y_1|a_0, t_0) \quad \gamma = \mathbb{P}(y_1|a_1, t_0) \quad Q = \frac{1 - \frac{\mathbb{P}(t_0|z_1)}{\epsilon}}{1 - \frac{\mathbb{P}(t_0|z_1)}{2\epsilon}} \quad \Phi = \frac{\epsilon + \frac{\tau}{2} - 1}{\epsilon + \frac{\tau}{2} - \frac{1}{2}} \quad \epsilon = \mathbb{P}(t_1)
$$

$$
\beta = \mathbb{P}(y_1|a_0, t_1) \quad \delta = \mathbb{P}(y_1|a_1, t_1) \quad R = \frac{1 - \frac{\mathbb{P}(t_1|z_0)}{1-\epsilon}}{1 - \frac{\mathbb{P}(t_1|z_0)}{2-2\epsilon}} \quad \Psi = \frac{1-\tau}{\tau} \quad \tau = \mathbb{P}(t_0|a_0)
$$

$\square$

## B.7 Proof of Theorem ??

*Proof.*

$$
\begin{aligned}
MeasBias(Y, A) &= \beta_{ya.t} - \beta_{ya.z} \\
&= \frac{\sigma_t^2 \sigma_{ya} - \sigma_{yt}\sigma_{ta}}{\sigma_a^2 \sigma_t^2 - \sigma_{ta}^2} - \frac{\sigma_z^2 \sigma_{ya} - \sigma_{yz}\sigma_{za}}{\sigma_a^2 \sigma_z^2 - \sigma_{za}^2} \\
&= \frac{\sigma_t^2(\sigma_a^2\alpha + \sigma_z^2\beta\gamma) - (\sigma_z^2\gamma\lambda + \sigma_z^2\alpha\beta\lambda)(\sigma_z^2\lambda\beta)}{\sigma_a^2\sigma_t^2 - \sigma_z^4\lambda^2\beta^2} - \alpha \\
&= \frac{\sigma_t^2\sigma_a^2\alpha + \sigma_t^2\sigma_z^2\beta\gamma - \sigma_z^4\gamma\lambda^2\beta - \sigma_z^4\lambda^2\beta^2\alpha}{\sigma_a^2\sigma_t^2 - \sigma_z^4\lambda^2\beta^2} \\
&= \frac{\alpha\,(\sigma_t^2\sigma_a^2 - \sigma_z^2\lambda^2\beta^2)}{\sigma_a^2\sigma_t^2 - \sigma_z^4\lambda^2\beta^2} + \frac{\sigma_t^2\sigma_z^2\beta\gamma - \sigma_z^4\gamma\lambda^2\beta}{\sigma_a^2\sigma_t^2 - \sigma_z^4\lambda^2\beta^2} - \alpha \\
&= \frac{\sigma_z{}^2\beta\gamma(\sigma_t{}^2 - \sigma_z{}^2\lambda^2)}{\sigma_a{}^2\sigma_t{}^2 - \sigma_z{}^4\lambda^2\beta^2}
\end{aligned}
\tag{57}
$$

In step (57), $\beta_{ya.z}$ is replaced by $\alpha$ (see proof of Theorem 3.2). $\square$

### B.7.1 Binary model, Intersectional Sensitive Variable

Interaction bias takes place in the presence of two sensitive attributes when the value of one sensitive attribute influences the effect of the other sensitive attribute on the outcome. Interaction bias is graphically illustrated in Figure 8. Note that regular DAGs are not able to express interaction. For this reason, we are employing the graphical representation proposed by Weinberg (2007). The arrows pointing to arrows, instead of nodes account for the interaction term. In a binary model interaction bias coincides with interaction term ($Interaction$) in the case of an intersectional sensitive attribute. Interaction bias also affects the individual measurement of the effect of sensitive attribute $A$ or $B$.

Given binary sensitive variables $A$, $B$ and a binary outcome $Y$, the joint discriminatio of $A = 0$ and $B = 0$ with respect to $Y$ can be expressed as follows:

$$
StatDisp(Y, A, B) = P(Y_1|a_1, b_1) - P(Y_1|a_0, b_0)
\tag{58}
$$

Here $Y = 1$ is a positive outcome, $A = 1$ and $B = 1$ represent the disadvantaged group.

**Theorem B.6.** *Under the assumption of no common parent for $A$ and $Y$ and $B$ and $Y$ [14] we can express $StatDisp(Y, A, B)$ in terms of causal effects of $A$ and $B$ and interaction between $A$ and $B$ on the additive scale:*

$$
\begin{aligned}
StatDisp(Y, A, B) = \ & \big[P(Y_1|a_1, b_0) - P(Y_1|a_0, b_0)\big] \\
& + \big[P(Y_1|a_0, b_1) - P(Y_1|a_0, b_0)\big] \\
& + Interaction(A, B)
\end{aligned}
$$

---

[14]This assumption is relatively easy to satisfy in case of immutable sensitive attributes such as gender or race because they are unlikely to have external causes. It is important to control for possible confounders when sensitive attributes can have external causes, for example, political beliefs can be influenced by education.

where $Interaction(A, B) = P(Y_1|a_1, b_1) - P(Y_1|a_0, b_1) - P(Y_1|a_1, b_0) + P(Y_1|a_0, b_0)$

Notice that: $P(Y_1|a_1, b_0) - P(Y_1|a_0, b_0)$ is the effect of $A$ on $Y$ in case there is no interaction, and similarly for $B$: $P(Y_1|a_0, b_1) - P(Y_1|a_0, b_0)$ is the effect of $B$ on $Y$ in case there is no interaction. To avoid confusion, we denote such expressions as $SD_{\cancel{Int}}(Y, A)$ and $SD_{\cancel{Int}}(Y, B)$ respectively.

## B.8   PROOF OF THEOREM 6.1

Given binary sensitive variables $A$, $B$ and a binary outcome $Y$, the discrimination with respect to only $A$ (and similarly for $B$) wrt to $Y$ can be expressed as follows:

$$StatDisp(Y, A) = P(Y_1|a_1) - P(Y_1|a_0) \tag{59}$$

**Theorem B.7.** *Under previously introduced assumption of no confounding, the discrimination with respect to $A$ can be decomposed into an interaction free discrimination and the interaction between $A$ and $B$:*

$$
\begin{aligned}
StatDisp(Y, A) = \big[ P(Y_1|a_1, b_0) - P(Y_1|a_0, b_0) \big] \\
+ P(b_1)Interaction(A, B)
\end{aligned}
\tag{60}
$$

$StatDisp(Y, B)$ can be decomposed in a similar way.

The interaction bias $IntBias(Y, A)$ is then defined as:

$$
\begin{aligned}
IntBias(Y, A) &= StatDisp(Y, A) - SD_{\cancel{Int}}(Y, A) \\
&= P(b_1)Interaction(A, B)
\end{aligned}
\tag{61}
$$

$IntBias(Y, B)$ is defined similarly:

$$
\begin{aligned}
IntBias(Y, B) &= StatDisp(Y, B) - SD_{\cancel{Int}}(Y, B) \\
&= P(a_1)Interaction(A, B)
\end{aligned}
\tag{62}
$$

*Proof.*

$$
\begin{aligned}
StatDisp(Y, A) &= \mathbb{P}(Y_1|a_1) - \mathbb{P}(Y_1|a_0) \\
&= \sum_b \mathbb{P}(Y_1|a_1, b)\mathbb{P}(b|a_1) - \sum_b \mathbb{P}(Y_1|a_0, b)\mathbb{P}(b|a_0) \\
&= \mathbb{P}(Y_1|a_1, b_1)\mathbb{P}(b_1|a_1) + \mathbb{P}(Y_1|a_1, b_0)\mathbb{P}(b_0|a_1) \\
&\quad - \mathbb{P}(Y_1|a_0, b_1)\mathbb{P}(b_1|a_0) - \mathbb{P}(Y_1|a_0, b_0)\mathbb{P}(b_0|a_0) \\
&= \mathbb{P}(Y_1|a_1, b_1)\mathbb{P}(b_1|a_1) + \mathbb{P}(Y_1|a_1, b_0)\mathbb{P}(1 - \mathbb{P}(b_1|a_1) \\
&\quad - \mathbb{P}(Y_1|a_0, b_1)\mathbb{P}(b_1|a_0) - \mathbb{P}(Y_1|a_0, b_0)\mathbb{P}(1 - \mathbb{P}(b_1|a_0) \\
&= \mathbb{P}(b_1|a_1)\big(\mathbb{P}(Y_1|a_1, b_1) - \mathbb{P}(Y_1|a_1, b_0)\big) + \mathbb{P}(Y_1|a_1, b_0) \\
&\quad + \mathbb{P}(b_1|a_0)\big(\mathbb{P}(Y_1|a_0, b_0) - \mathbb{P}(Y_1|a_0, b_1)\big) - \mathbb{P}(Y_1|a_0, b_0)
\end{aligned}
$$

Since $A$ and $B$ are independent $\mathbb{P}(b_1|a_1) = \mathbb{P}(b_1|a_0) = \mathbb{P}(b_1)$. It follows that:

$$
\begin{aligned}
StatDisp(Y, A) &= \mathbb{P}(b_1)Int(A, B) + \mathbb{P}(Y_1|a_1, b_0) - \mathbb{P}(Y_1|a_0, b_0) \\
&= \mathbb{P}(b_1)Int(A, B) + CDE_b(Y, A)
\end{aligned}
$$

And the interaction bias is:

$$IntBias(Y, A) = StatDisp(Y, A) - CDE_b(Y, A)$$
$$= \mathbb{P}(b_1)Int(A, B)$$

□

## C CAUSAL GRAPHS

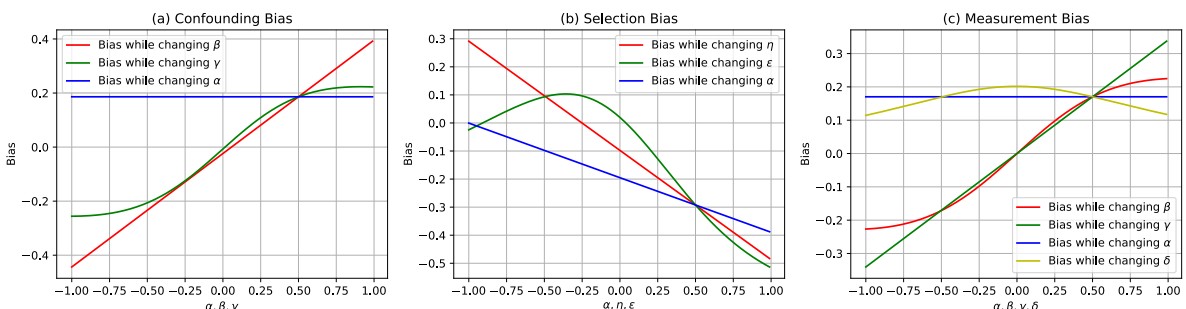

Figure 18: Bias Magnitude while changing one variable and holding the other variables at $0.5$.

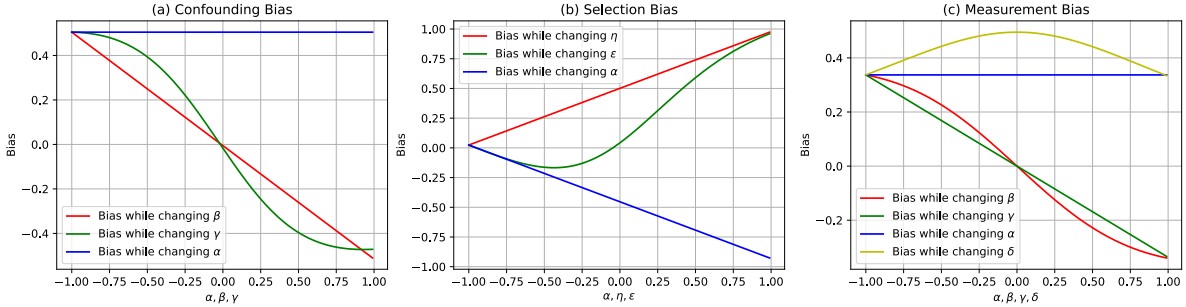

Figure 19: Bias Magnitude while changing one variable and holding the other variables at $-1.0$.

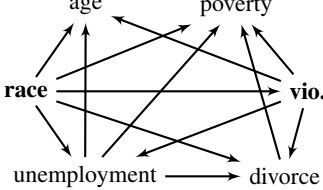

Figure 20: The graph for the communities and crime dataset. 'divorce', 'age', 'poverty' and 'un-employement' are the colliders between 'race' and 'violence' (vio.). The graph is produced using LiNGAM algorithm.

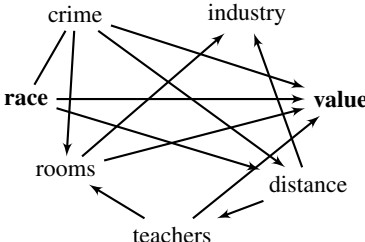

Figure 21: The graph for the Boston housing data set. 'Crime' is a possible confounder between 'race' and 'value'.The graph is produced using GES algorithm.

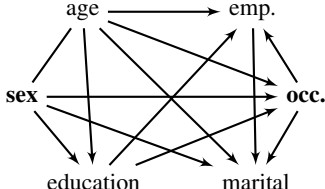

Figure 22: The graph for the Dutch data set. 'Marital Status' is a collider between 'sex' and 'occupation' (occ.). The graph is produced using GES algorithm.

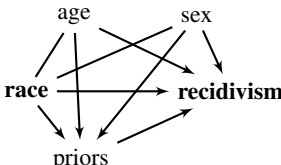

Figure 23: The graph for the Compas dataset. 'Age' and 'sex' are possible confounders between 'race' and 'recidivism'. The graph is produced using PC algorithm.

## D    GENERATION OF SYNTHETIC DATA

### D.1    MEASUREMENT BIAS CATA

The variables $Z$, $A$, $T$, and $Y$ are binary Bernoulli variables controlled by the parameter $p_1$. Conditional dependencies of the measurement bias structure define how the parameter $p_1$ depends on the value of the parent variables.

$$Z \sim (p) = \begin{cases} p_1, \\ p_0 = 1 - p_1 \end{cases} \quad (63)$$

$$A \sim (Z; p) = \begin{cases} p_1, & \text{if } Z = 1, \\ p_0 = 1 - p_1 \\ p_1', & \text{if } Z = 0. \\ p_0 = 1 - p_1' \end{cases} \quad (64)$$

$$T \sim (Z; p) = \begin{cases} p_1, & \text{if } Z = 1, \\ p_0 = 1 - p_1 \\ p_1', & \text{if } Z = 0. \\ p_0' = 1 - p_1' \end{cases} \quad (65)$$

$$Y \sim (p; Z, A) = \begin{cases} p_1 = 0.5 * z + 0.5 * a, \\ p_0 = 1 - p_1 \end{cases} \quad (66)$$

The parameters $p_1$, $p_0$, $p_1'$ and $p_0'$ are generated randomly and take value between $0$ and $1$.

## D.2 LINEAR DATA

To analyze the different types of bias in the linear case, we generate synthetic data according to the following models. Without loss of generality, the range of possible values of all coefficients ($\alpha, \beta, \gamma, \eta, \epsilon$, and $\delta$) is $[-1.0, 1.0]$

*Confounding Structure:*

$Z = \mathcal{U}_z,$
$A = \beta Z + \mathcal{U}_a,$
$Y = \alpha A + \gamma Z + \mathcal{U}_y$

*Colliding Structure:*

$A = \mathcal{U}_a,$
$Y = \alpha A + \mathcal{U}_y,$
$W = \eta A + \epsilon Y + \mathcal{U}_w$

*Measurement Structure:*

$Z = \mathcal{U}_z,$
$A = \beta Z + \mathcal{U}_a,$
$Y = \alpha A + \gamma Z + \mathcal{U}_y,$
$T = \delta Z + \mathcal{U}_t$

$\mathcal{U}_z \sim \mathcal{N}(0,1),$
$\mathcal{U}_a \sim \mathcal{N}(0,1),$
$\mathcal{U}_y \sim \mathcal{N}(0,1),$
$\mathcal{U}_w \sim \mathcal{N}(0,1),$
$\mathcal{U}_t \sim \mathcal{N}(0,1).$

## E CONCURRENT BIASES

**Confounding and selection biases.** In presence of one or several confounder and collider variables, the estimation of discrimination can suffer from both confounding and selection biases simultaneously. Figure 24 shows the simplest case. According to Definitions B.1 and B.4, confounding bias can be isolated by adjusting on the confounder variable $ConfBias(Y, A) = StatDisp(Y, A) - StatDisp_Z(Y, A)$[15] ($\beta_{ya} - \beta_{ya.z}$ in the linear case), whereas selection bias can be isolated by cancelling the adjustment on the collider variable $SelBias(Y, A) = StatDisp(Y, A)_W - StatDisp(Y, A)$ ($\beta_{ya.w} - \beta_{ya}$ in the linear case). The total bias in presence of both types of bias can then be estimated as $StatDisp(Y, A)_W - StatDisp(Y, A)_Z$ in the binary case and $\beta_{ya.w} - \beta_{ya.z}$ in the linear case.

**Confounding and measurement biases.** Measurement bias (Figure 7) is defined as the difference in estimating $StatDisp$ when adjusting on the proxy variable ($T$) instead of the unobservable/unmeasurable confounder variable ($Z$). For the binary case, it corresponds to the difference $StatDisp_T(Y, A) - StatDisp_Z(Y, A)$. For the linear case, it corresponds to the difference between the partial regression coefficients $\beta_{ya.t} - \beta_{ya.z}$. The difference between the adjustment free estimation of $StatDisp(Y, A)$ (the regression coefficient $\beta_{ya}$ in the linear case) and $StatDisp_T(Y, A)$ ($\beta_{ya.t}$) corresponds to the total of both confounder and measurement biases.

**Selection and measurement biases.** Figure 25 shows the simplest case where measurement and selection biases occur simultaneously. Adjusting on both the proxy ($T$) and the collider ($W$) variables ($StatDisp_{TW}(Y, A)$ and $\beta_{ya.tw}$) leads to both types of biases occurring simultaneously. Substracting $StatDisp_Z(Y, A)$ (respectively $\beta_{ya}$) from $StatDisp_{TW}(Y, A)$ (respectively $\beta_{ya.tw}$) coincides with the sum of selection and measurement biases in the binary and linear cases respectively.

**Confounding, selection, and measurement biases.** In the same simple case of Figure 25, the difference between adjusting on variables $T$ and $W$ on one hand and adjusting on $Z$ on the other hand ($StatDisp_{TW}(Y, A) - StatDisp_Z(Y, A)$ in the binary case and $\beta_{ya.tw} - \beta_{ya.z}$ in the linear case) encompasses the three types of bias.

**Confounding and interaction biases.** In presence of interaction between two sensitive variables, confounding bias can be decomposed into interaction free portion and an interaction term. Figure 26 shows a simple confounding structure between $A$ and $Y$ and a second sensitive variable $B$ which is interacting with the effect of $A$ on $Y$. In the binary case, the confounding bias $ConfBias(Y, A)$ (Definition B.1) can be decomposed as follows:

**Proposition E.1.**

$$\begin{aligned} ConfBias(Y, A) &= StatDisp(Y, A) - StatDisp_Z(Y, A) \\ &= SD_{\cancel{Int}}(Y, A) - SD_{\cancel{Int}_Z}(Y, A) \\ &\quad + P(b_1)(Interaction(A, B) - Interaction_Z(A, B)) \end{aligned} \tag{67}$$

$$\tag{68}$$

---

[15]Notice that, by the backdoor formula, $StatDisp_Z(Y, A)$ coincides with $ACE(Y, A)$.

*where*

$$SD_{\cancel{Int}_Z}(Y, A) = \sum_Z (P(y_1|a_1, b_0, z) - P(y_1|a_0, b_0, z))P(z)$$

$$Interaction_Z(A, B) = \sum_Z \big( P(y_1|a_1, b_1, z) - P(y_1|a_0, b_1, z) \\ - P(y_1|a_1, b_0, z) + P(y_1|a_0, b_0, z) \big) P(z)$$

In the same example of Figure 26, the confounding bias in case of intersectionality (two interacting sensitive variables) can be decomposed as follows:

**Proposition E.2.**

$$ConfBias(Y, A, B) = StatDisp(Y, A, B) - StatDisp_Z(Y, A, B) \\ = SD_{\cancel{Int}}(Y, A) - SD_{\cancel{Int}_Z}(Y, A) \\ + Interaction(A, B) - Interaction_Z(A, B) \quad (69)$$

$$(70)$$

In the slightly different structure where $Z$ is also a confounder between $B$ and $Y$ (Figure 27), the term $SD_{\cancel{Int}}(Y, B) - SD_{\cancel{Int}_Z}(Y, B)$ needs to be added to the $ConfBias(Y, A, B)$ expression above.

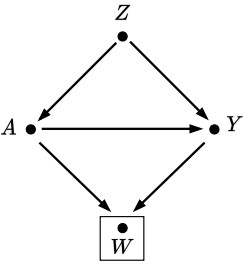

Figure 24: Confounding and colliding bias

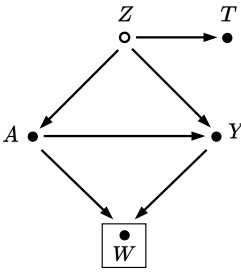

Figure 25: Confounding, colliding, and measurement bias

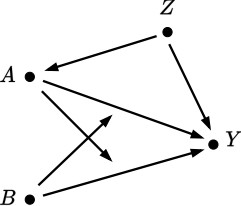

Figure 26: Interaction and confounding bias

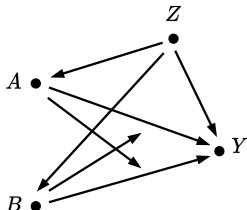

Figure 27: Interaction and confounding bias