# OpenReview forum: "Dissecting Causal Biases"
_ICLR.cc/2024/Conference — ICLR 2024 Conference Withdrawn Submission_

### Official Review · Reviewer_tMUj · 2023-10-27

**Soundness:** 3 good
**Presentation:** 3 good
**Contribution:** 1 poor
**Rating:** 3
**Confidence:** 3

**Summary:**

The paper explores "causal biases" in machine learning decision systems, arising from data collection methods. Using causality tools, it defines and analyzes four key bias sources: confounding, selection, measurement, and interaction. Through closed-form expressions, the study evaluates when these biases are most pronounced or absent.

**Strengths:**

- The problem is well motivated and the writing of the paper is good
- The idea of distinguishing between discrimination and bias is (to the best of my knoweledge) novel  and important

**Weaknesses:**

Overall, it is not clear to me what is learned from the paper. While the closed-form solutions are novel, they can be easily derived using existing causal inference tools. Further, there is little evidence supporting why these results are useful. Experiments could have strengthened this case, but the experimental section is toyish and it does not illustrate any real-world use-case. Lastly, the results in the paper crucially rely on the knoweldge of the causal structure, an assumption which is very unlikely to hold in practice, making the relevance of the paper for practicioners even more questionable. For these reasons, I'm not convinced that the paper makes an acceptance-worthy contribution.

**Questions:**

- It would have been good to illustrate a real-world use-case of the results in the paper. How can these results help us inform decisions in the real-world?

---

### Official Review · Reviewer_Ynmd · 2023-10-31

**Soundness:** 2 fair
**Presentation:** 2 fair
**Contribution:** 2 fair
**Rating:** 3
**Confidence:** 3

**Summary:**

The authors proposed a set of four biases present in data generating processes. The authors state that these biases come from a misconception of the causal mechanisms used to represent the used data (missing links, or relations between the variables). They addressed these graphical misconceptions by naming them and formalized them as closed form expressions. The provide theoretical support for their proposed formulae.

**Strengths:**

- [S1] Understanding different types of biases and using causal models to analyze and structure them, is an interesting approach that has the potential to make an interesting paper.
- [S2] Providing the causal graphs for the different types of biases supports the understanding of the differences between the different types of biases.

**Weaknesses:**

- [W1] The transitions between the different sections of the paper are not smooth. There's no clear outline. First, the four types of biases are presented, then they are explained in more detail (close forms), and then they are approached by the modeling and experimental perspective. This sequence is only clear after the paper is read. I would suggest a more extensive outline of the structure of the paper somewhere in the introduction so the reader knows what to expect. I would also suggest adding some motivation to support the existence of the different "macro"-sections (those mentioned in the previous sentences).
- [W2] The presentation and difference of biases, discrimination and measurements is not clear, which makes difficult to understand what the purpose of the work is.  In the introduction authors dedicated a full paragraph to make a distinction between discrimination and bias, but no clear connections are drawn between the two of them. This right after they motivate their work by saying "Addressing the problem of discrimination involves
two main tasks. First, measuring discrimination as accurately and reliably as possible. Second, mitigating discrimination." But a few lines below, the authors state that "The main contribution of the paper is to use tools and existing results from the field of causality to generate closed-form expressions of four sources of bias." Given this, a clear connection between biases and the measurement of discrimination is needed.
- [W3] The results are presented in a non cohesive way inside the more theoretical sections. Some theorems are presented without a clear motivation.
 [W4] The paper is not very connected nor consequent: the work focuses in presenting closed form expressions of the different types of bias. They back up the importance of these formulae by saying that measuring this in the data is crucial to mitigate discrimination. Once they reach the experimental part, they show results on these measurements but do not reference the actual equations used for this. They also do not try to show that the assumptions of the theorems are valid.
- [W5] One of the most important points that should support this work (experimental validation) is shown from Figure 10-15 but these plots are not easy to read. The text of the legends, labels etc. are very small.
- [W6] Concepts such as confounding bias, sampling bias and intersectionality have been explored. studied and discussed by prior work. Little connection is drawn to this. For example, in the introduction paragraph on confounding bias (page 2), there is no citation to prior work. Similar holds for sampling bias


Further comments:
- Much of the theoretical sections is built from the linear regression coefficients. I would suggest adding a more detailed description of it and how its implications before presenting those sections.
- In section 5, it is written "Consistent with previous work," with no citation shown.
- Theorem 5.1 introduces is built around an Eq. (11) where no equal sign is written, which seems an odd way of defining a quantity (MeasBias).
- In section 7, the intro of that section is rather confusing: "The aim is to identify the cases where a given estimation of discrimination is biased and at which...". The aim of the authors' work? The aim of that section?


-Typos:
-- Second paragraph of the introduction has an 'etc..' (missing a dot?).
-- ConfBias is presented in different fonts.
-- In addition, the considered causal structures *mots*? often show.

**Questions:**

[Q1] Theorem 3.1 defines the cofounder bias using quantities that were not mentioned before (up to that point Figure 5 is not referenced). Theorem 3.2 does reference this Figure but the formula makes use of delta which is not addressed in that Figure; it is only defined in Section 5. The other used quantities are defined there too (again). Are these new definitions or do they match the linear regression coefficients used in the definition of the CoefBias?

[Q2] Intersectionality, where for two sensitive attributes the "joint effect [is] smaller or greater than the sum of individual effects" (page 3), is an inherent non-linear concept. The authors do indeed only show results for the other three types of biases for linear case. Is the non-linearity that is inherent in interaction bias the reason, why intersectional (interaction) bias is excluded from the results in 7.2 Or is this not the case?

---

### Official Review · Reviewer_P3Ft · 2023-11-01

**Soundness:** 3 good
**Presentation:** 3 good
**Contribution:** 1 poor
**Rating:** 3
**Confidence:** 4

**Summary:**

This paper presents quantifications of various forms of causal biases (e.g., selection, confounding, interaction) under linearity.

**Strengths:**

See below for a contextual discussion of strengths and perceived weaknesses.

**Weaknesses:**

This paper is well written. It strikes me as more of a review piece summarizing existing literature or approaches than a stand-alone independent research contribution (e.g., the derivations under linearity are relatively standard; the class of biases identified are in general well-recognized within the causal inference community). In it's current framing, I think the paper has a place in the literature, but perhaps at a different venue (perhaps something like the Journal of Causal Inference, where review pieces are more typical, although I appreciate the motivation of the paper to "bring causal inference ideas to model training" [my summary]). The paper could also bolster its independent contribution by further focusing on the interaction between bias types (e.g., how selection and confounding bias interrelate under linearity).

I have a few more specific comments.

(1) The examples in Figure 1-4 are helpful, although the selected examples of the different bias types don't always feel the most natural (i.e., would have to be 'argued for' in the sense that different DAGs seem relevant (i.e., in certain situations it could be that labor union activism affects political beliefs through information channels, etc.).

(2) The paper is framed partly in terms of "let's bring causal inference ideas to model training" [my summary]. This is a reasonable framing, although the examples and the analyses sometimes don't focus as much on model training as much as generic causal inference questions. Therefore, I as a reader was left somewhat confused about the contribution (i.e., the abstract suggested more of an emphasis on the "causality for ML training" aspect, but the text seemed to focus on causal biases in a more generic (and therefore perhaps less novel) sense.

Overall, this paper is useful in its succinct and clear summarization of various causal biases, but has limitations in terms of broader contribution and framing.

**Questions:**

I do not have major questions at this time, but will be sure to ask in followup discussions if they come up.

---

### Official Review · Reviewer_dVLJ · 2023-11-01

**Soundness:** 3 good
**Presentation:** 3 good
**Contribution:** 4 excellent
**Rating:** 6
**Confidence:** 3

**Summary:**

This paper categorize four types of causal biases in the data generating process. The authors exemplify these causal biases with a hiring examples along with the corresponding causal graphs. Then for each type of the causal biases, the authors proposed a formal expression and derived the closed-forms in the linear regressions case. Finally, on the linear synthetic data, the authors simulate the magnitude of different type of biases, and discuss the conditions under which the bias terms can be canceled out. The authors also run experiments with 5 real-world datasets to illustrate the behavior of causal biases.

**Strengths:**

I like the scope of this paper that can help to understand the origins of biases from the data generation process. The category of biases through the lens of causality seems to be novel and interesting. I appreciate the authors use the detailed causal graph and the story of hiring to help illustrate the implications for each causal biases.

**Weaknesses:**

There are a couple of minor typos. Figure 5 is not referenced in Theorem 3.1 so that readers may be confused about what are the variables of $\alpha$, $\beta$, $\gamma$. And please see [Q1] for detailed questions. Fonts in Figure 10-Figure 15 are too small and hard to recognize.

**Questions:**

[Q1] What are the variables of $\alpha$, $\beta$, $\gamma$ and $\delta$? Especially, the variable $\delta$ does not appeared in the Figure 5.